# The sheddase ADAM10 is a potent modulator of prion disease

**Hermann C Altmeppen[1], Johannes Prox[2], Susanne Krasemann[1], Berta Puig[1], Katharina Kruszewski[3], Frank Dohler[1], Christian Bernreuther[1], Ana Hoxha[1], Luise Linsenmeier[1], Beata Sikorska[4], Pawel P Liberski[4], Udo Bartsch[3], Paul Saftig[2†], Markus Glatzel[1\*†]**

[1]Institute of Neuropathology, University Medical Center Hamburg-Eppendorf, Hamburg, Germany; [2]Institute of Biochemistry, Christian Albrechts University, Kiel, Germany; [3]Department of Ophthalmology, University Medical Center Hamburg-Eppendorf, Hamburg, Germany; [4]Department of Molecular Pathology and Neuropathology, Medical University Lodz, Lodz, Poland

**Abstract** The prion protein (PrP$^C$) is highly expressed in the nervous system and critically involved in prion diseases where it misfolds into pathogenic PrP$^{Sc}$. Moreover, it has been suggested as a receptor mediating neurotoxicity in common neurodegenerative proteinopathies such as Alzheimer's disease. PrP$^C$ is shed at the plasma membrane by the metalloprotease ADAM10, yet the impact of this on prion disease remains enigmatic. Employing conditional knockout mice, we show that depletion of ADAM10 in forebrain neurons leads to posttranslational increase of PrP$^C$ levels. Upon prion infection of these mice, clinical, biochemical, and morphological data reveal that lack of ADAM10 significantly reduces incubation times and increases PrP$^{Sc}$ formation. In contrast, spatiotemporal analysis indicates that absence of shedding impairs spread of prion pathology. Our data support a dual role for ADAM10-mediated shedding and highlight the role of proteolytic processing in prion disease.

**\*For correspondence:**
m.glatzel@uke.de

†These authors contributed equally to this work

**Competing interests:** The authors declare that no competing interests exist.

**Reviewing editor**: Bart De Strooper, VIB Center for the Biology of Disease, KU Leuven, Belgium

## Introduction

The cellular prion protein (PrP$^C$) is a glycosylphosphatidylinositol (GPI)-anchored lipid raft constituent highly expressed in neurons of the central nervous system (CNS). Misfolding of PrP$^C$ into its pathogenic isoform, PrP$^{Sc}$, occurs by a self-perpetuating process of templated conformational conversion and causes transmissible and invariably fatal prion diseases (*Prusiner, 1982*). In these diseases, PrP$^C$ not only represents the substrate for conversion but also a prerequisite of prion-associated neurotoxicity (*Büeler et al., 1993*; *Brandner et al., 1996*; *Mallucci et al., 2003*). In contrast to PrP$^C$, PrP$^{Sc}$ is prone to aggregation and is characterized by a high β-sheet content and partial resistance to digestion with proteinase K (PK). Moreover, PrP$^{Sc}$ constitutes an essential component of infectious prion particles or 'prions' (*Prusiner, 1982*; *Silveira et al., 2005*).

Recently, a central role in more common proteinopathies, such as Alzheimer's disease (AD), has been attributed to PrP$^C$ where it was shown to act as a neuronal receptor for neurotoxic amyloid-β (Aβ) oligomers (*Lauren et al., 2009*; *Dohler et al., 2014*) and a variety of other disease-associated β-sheet rich protein assemblies including PrP$^{Sc}$ (*Resenberger et al., 2011*). Despite some degree of controversy regarding this receptor function and its relevance in AD (*Balducci et al., 2010*; *Benilova and De Strooper, 2010*; *Calella et al., 2010*; *Kessels et al., 2010*), binding of pathogenic oligomers to the flexible N-terminus of PrP$^C$ is thought to initiate a cascade of events that mediates their neurotoxicity and results in synaptic degeneration and neuronal loss (*Resenberger et al., 2011*; *Larson et al., 2012*; *Um et al., 2012*).

**eLife digest** Prion proteins are anchored to the surface of brain cells called neurons. Normally, prion proteins are folded into a specific three-dimensional shape that enables them to carry out their normal roles in the brain. However, they can be misfolded into a different shape known as PrP^Sc, which can cause Creutzfeldt-Jakob disease and other serious conditions that affect brain function and ultimately lead to death.

The PrP^Sc proteins can force normal prion proteins to change into the PrP^Sc form, so that over time this form accumulates in the brain. They are essential components of infectious particles termed 'prions' and this is why prion diseases are infectious: if prions from one individual enter the brain of another individual they can cause disease in the recipient. The UK outbreak of variant Creutzfeldt-Jakob disease in humans in the 1990s is thought to be due to the consumption of meat from cattle with a prion disease known as mad cow disease.

An enzyme called ADAM10 can cut normal prion proteins from the surface of neurons. However, it is not clear whether ADAM10 can also target the PrP^Sc proteins and what impact this may have on the development of prion diseases.

Here, Altmeppen et al. studied mutant mice that were missing ADAM10 in neurons in the front portion of their brain. These mice had a higher number of normal prion proteins on the surface of their neurons than normal mice did. When mice missing ADAM10 were infected with prions, more PrP^Sc accumulated in their brain and disease symptoms developed sooner than when normal mice were infected. This supports the view that mice with higher numbers of prion proteins are more vulnerable to prion disease. However, disease symptoms did not spread as quickly to other parts of the brain in the mice missing ADAM10. This suggests that by releasing prion proteins from the surface of neurons, ADAM10 helps PrP^Sc proteins to spread around the brain.

Recently, it has been suggested that prion proteins may also play a role in Alzheimer's disease and other neurodegenerative conditions. Therefore, Altmeppen et al.'s findings may help to develop new therapies for other forms of dementia. The next challenge is to understand the precise details of how ADAM10 works.

PrP^C is subject to two conserved proteolytic cleavage events under physiological conditions, termed α-cleavage and shedding (*Borchelt et al., 1993*; *Harris et al., 1993*; *Chen et al., 1995*; *Mange et al., 2004*). These cleavages likely influence physiological functions of PrP^C (*Aguzzi et al., 2008*; *Linden et al., 2008*) and, importantly, its role in neurodegenerative diseases (reviewed in *Altmeppen et al., 2013*). While α-cleavage, occurring in the middle of the PrP^C sequence and producing a soluble N1 and a membrane-attached C1 fragment, confers neuroprotection with regard to prion diseases (*Lewis et al., 2009*; *Westergard et al., 2011*; *Turnbaugh et al., 2012*; *Campbell et al., 2013*) and Aβ-associated neurotoxicity (*Guillot-Sestier et al., 2009*; *Resenberger et al., 2011*; *Beland et al., 2012*; *Guillot-Sestier et al., 2012*; *Fluharty et al., 2013*), the role of an extreme C-terminal cleavage in close proximity to the plasma membrane, termed shedding, remains enigmatic. On the one hand, by reducing the substrate for prion conversion and the receptor for neurotoxicity, this proteolytic release of almost full length PrP^C from the plasma membrane could be protective with regard to prion diseases (*Marella et al., 2002*; *Heiseke et al., 2008*; *Altmeppen et al., 2012*). Moreover, recombinant or transgenically expressed anchorless PrP^C mimicking shed PrP^C has been shown to antagonize both PrP^Sc propagation and Aβ neurotoxicity, respectively, thus indicating a protective activity by blocking toxic conformers and preventing their access to the cell (*Meier et al., 2003*; *Calella et al., 2010*; *Nieznanski et al., 2012*; *Fluharty et al., 2013*; *Yuan et al., 2013*). On the other hand, shedding could accelerate an ongoing prion disease by increasing production and subsequent spread of anchorless prions within the CNS. In fact, PrP^Sc can be shed from the cellular surface in vitro (*Taylor et al., 2009*) and anchorless PrP^C can, in principle, be converted to PrP^Sc, resulting in an altered type of prion disease in transgenic mice (*Chesebro et al., 2005*, *2010*). High expression of anchorless PrP^C leads to formation of prions and a late onset neurological disease (*Stöhr et al., 2011*).

While the in vivo identity of the protease(s) responsible for the α-cleavage of PrP^C remains a matter of controversy (*Vincent et al., 2001*; *Tveit et al., 2005*; *Taylor et al., 2009*; *Oliveira-Martins et al., 2010*;

*Altmeppen et al., 2011, 2012*; *Beland et al., 2012*; *Liang et al., 2012*; *Wik et al., 2012*; *Mays et al., 2014*; *McDonald et al., 2014*), we recently showed that a disintegrin and metalloproteinase ADAM10 is the relevant sheddase of PrP$^C$ in vivo (*Altmeppen et al., 2011*), as previously suggested (*Parkin et al., 2004*; *Taylor et al., 2009*) and subsequently confirmed (*Wik et al., 2012*; *Ostapchenko et al., 2013*; *McDonald et al., 2014*) by in vitro experiments of others. Using conditional $^{Nestin}$ADAM10 knockout mice (*Jorissen et al., 2010*), we recently showed that depletion of the protease in neural precursors abolishes shedding and leads to posttranslational accumulation of PrP$^C$ in the early secretory pathway indicative of a regulatory role of ADAM10 in PrP$^C$ membrane homeostasis (*Altmeppen et al., 2011*). However, due to the perinatal lethality of these mice, the influence of ADAM10 on the course of prion disease remained unsolved. In this study we used novel viable ADAM10 conditional knockout (ADAM10 cKO or A10 cKO) mice with specific postnatal deletion of the protease in forebrain neurons (*Prox et al., 2013*). We show that lack of ADAM10 leads to (i) elevated (membrane) levels of PrP$^C$, (ii) drastically shortened incubation times of prion disease, (iii) increased prion conversion, and (iv) upregulation of calpain levels. In contrast, spread of prion-associated pathology within the brain was reduced. It is intriguing that the absence of a protease involved in the constitutive processing of PrP$^C$ significantly impacts the course of prion disease, thus enforcing the relevance of proteolytic processing events in neurodegeneration.

## Results

### Increased (membrane) levels and lack of shedding of PrP$^C$ in different cellular models of ADAM10 depletion

Maintenance of plasma membrane levels of PrP$^C$ by ADAM10-mediated shedding may represent a conserved mechanism in different cellular lineages. To further investigate this we generated neural stem cells (NSCs) from embryonic day (E) 14 wild-type and A10 cKO mice with inactivation of the *Adam10* gene in neuroectodermal progenitor cells ($^{Nestin}$A10 KO mice) (*Jorissen et al., 2010*). As shown in *Figure 1A*, surface biotinylation experiments on neuronally differentiated NSCs revealed that membrane levels of PrP$^C$ were increased 1.56-fold ($\pm 0.12$; SEM) in the absence of ADAM10 (n = 9 independent samples) compared with wild-type controls (set to $1 \pm 0.13$; n = 9). Moreover, genetic reintroduction of *Adam10* into NSC cultures of $^{Nestin}$A10 KO mice was sufficient to reduce membrane levels of PrP$^C$ ($0.95 \pm 0.11$; n = 8) and thus to restore physiological wild-type conditions. Nucleofection of $^{Nestin}$A10 KO cells with a vector lacking the *Adam10* cDNA did not show any effect on PrP$^C$ membrane levels ($1.55 \pm 0.18$; n = 5). Indirect immunofluorescence analyses of non-permeabilized neuronally differentiated NSCs confirmed the biochemical results by showing increased intensity of PrP$^C$ surface immunostaining in $^{Nestin}$A10 KO cells and $^{Nestin}$A10 KO cells nucleofected with a control vector compared with wild-type control cells, and a similar intensity of PrP$^C$ surface immunostaining in A10-nucleofected $^{Nestin}$A10 KO and control cells (*Figure 1B*).

In addition, we analyzed murine embryonic fibroblasts (MEFs) derived from mice with a complete knockout of ADAM10 (*Hartmann et al., 2002*) with regard to PrP$^C$ levels (*Figure 1C and D*). As expected, we found increased total PrP$^C$ levels in ADAM10 knockout MEFs by (i) immunofluorescence analysis of permeabilized cells (*Figure 1C*, upper part) and (ii) Western blot assessment of MEF lysates (*Figure 1D*, left part). In non-permeabilized wild-type MEFs, colocalization could be observed between the protease ADAM10 and its substrate PrP$^C$ at the plasma membrane (*Figure 1C*, bottom row). Next, we directly investigated the shedding of PrP$^C$ in ADAM10 knockout and wild-type MEFs by biochemical analysis of culture supernatants. Results obtained with concentrated media and with immunoprecipitation of shed PrP$^C$ from media showed that shedding is impaired in ADAM10 knockout MEFs compared with wild-type MEFs (*Figure 1D*, right part).

Finally, we assessed PrP$^C$ levels and the shedding of PrP$^C$ in primary neurons of $^{Nestin}$A10 KO and wild-type control mice as well as in mice deficient for PrP$^C$ (*Prnp$^{0/0}$*) and in PrP$^C$ overexpressing *Tg(Prnp)a20* mice (known and hereafter referred to as *tga20*) (*Figure 1E*). Confirming our previous study (*Altmeppen et al., 2011*), shedding of PrP$^C$ was absent in $^{Nestin}$A10 KO neurons while *tga20* neurons showed increased levels of shed PrP$^C$ compared with wild-type controls.

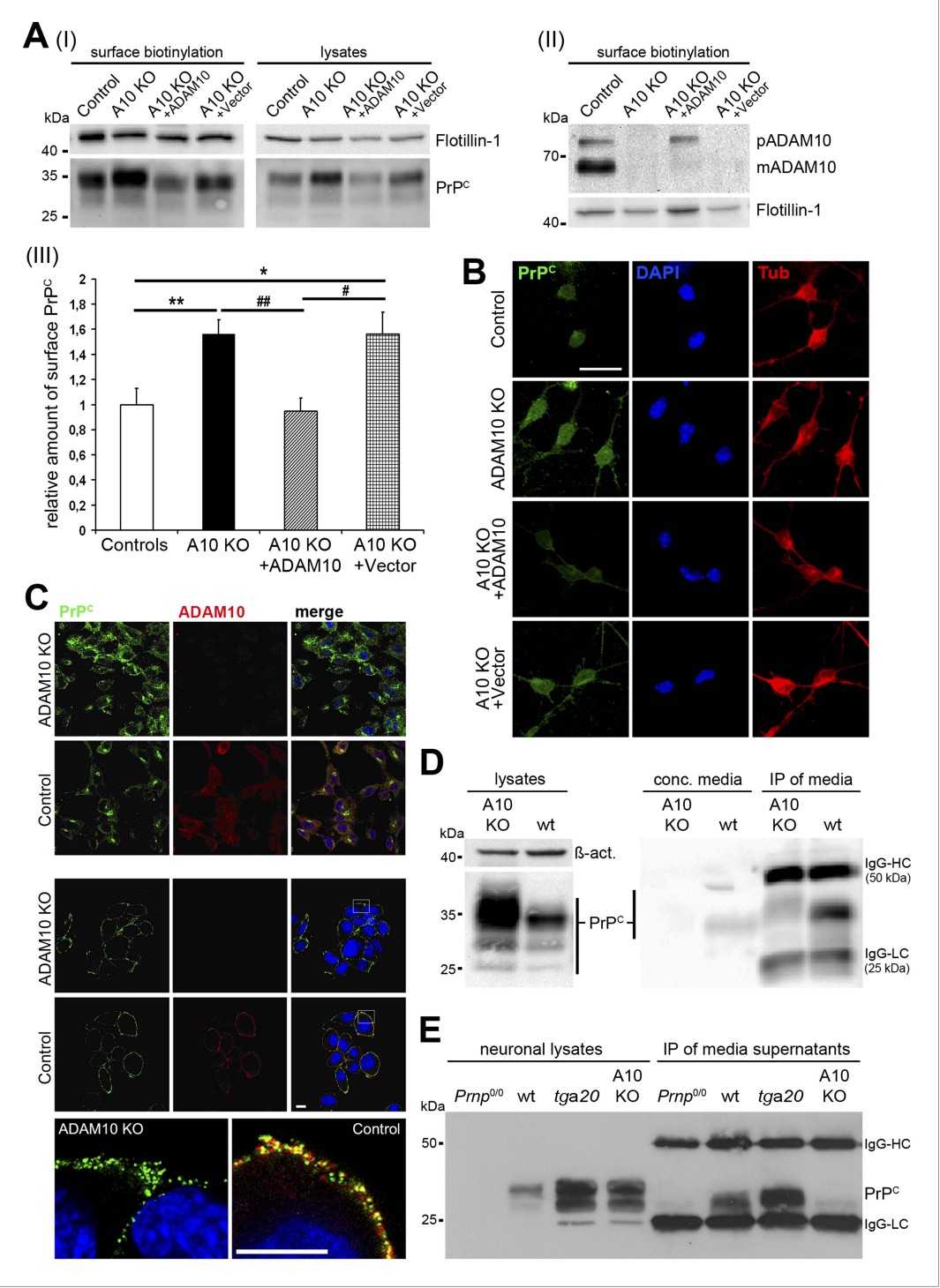

**Figure 1**. Characterization of PrP$^C$ levels in different cellular models of ADAM10 deficiency. (**A**) Representative Western blots showing membrane levels of PrP$^C$ as revealed by surface biotinylation (I; on the left) and total PrP$^C$ levels in lysates (I; on the right) as well as ADAM10 surface expression (II) of neuronally differentiated neural stem cells (NSCs) from $^{Nestin}$A10 KO and littermate control mice and after genetic reintroduction of *Adam10* (A10 KO + ADAM10) or nucleofection with control vector (A10 KO + Vector) into NSCs of $^{Nestin}$A10 KO mice. Flotillin served as loading control. (III) Quantification of densitometric analysis of PrP$^C$ membrane levels of experimental groups mentioned above (n = 9 independent samples for controls [set to 1]; n = 9 for $^{Nestin}$A10 KO; n = 8 for $^{Nestin}$A10 KO + ADAM10; n = 5 for $^{Nestin}$A10 KO + Vector; significance: **p = 0.0054; ##p = 0.0014; *p = 0.0336 ; #p = 0.0212). Error bars indicate SEM. (**B**) Representative immunofluorescent PrP$^C$ (green) surface staining of neuronally differentiated

*Figure 1. continued on next page*

*Figure 1. Continued*

NSCs derived from ^(Nestin)A10 KO (without [second row] or with genetic reintroduction of ADAM10 [third row] or vector only [fourth row]) and littermate control mice (first row), respectively. Tubulin (red) was stained after permeabilization of cells to confirm neuronal differentiation of NSCs. DAPI (blue) marks nuclei. (**C**) Representative immunostaining of PrP^C (green) and ADAM10 (red) in permeabilized (upper two rows) and non-permeabilized (lower three rows) murine embryonic fibroblasts (MEFs) derived from mice with a complete knockout of ADAM10 (ADAM10 KO) or wild-type mice (control). Higher resolution of white boxes is shown in the bottom row and reveals colocalization of PrP^C and ADAM10 at the plasma membrane of wild-type control MEFs. Scale bars in **B** and **C** represent 10 µm. (**D**) Western blot analysis of cell-associated PrP^C levels in ADAM10 knockout (A10 KO) and wild-type (wt) MEF lysates (left part: actin served as loading control). Levels of shed PrP^C were assessed in cell culture media supernatants of ADAM10 knockout and wild-type MEFs by filter column concentration (conc. media) and immunoprecipitation (IP) with a PrP^C-specific antibody respectively (right part). (**E**) Levels of cell-associated (neuronal lysates) and shed PrP^C (IP of media supernatants) in primary neuronal cultures of prion protein knockout (*Prnp*^0/0), wild-type (wt; C57BL/6), prion protein overexpressing (*tga20*), and ^(Nestin)A10 KO mice at embryonic day 14. IgG-HC and IgG-LC mark signals for heavy and light chain of the capturing antibody POM2.

Taken together, data obtained from different murine cellular models having a deletion of *Adam10* confirmed the role of this protease as the functionally relevant sheddase of PrP^C and, thus, as a regulator of PrP^C membrane homeostasis.

## Lack of ADAM10 in neurons of the forebrain results in increased levels of PrP^C

Using conditional ^(Nestin)A10 knockout mice, we previously showed that lack of ADAM10-mediated shedding leads to increased neuronal levels of PrP^C (*Altmeppen et al., 2011*). However, due to the perinatal lethality of these mice we were unable to investigate the impact of ADAM10 deficiency on the course of prion disease. Therefore, new conditional ^(Camk2a)ADAM10 knockout mice (ADAM10 cKO or A10 cKO) lacking Adam10 in neurons of the forebrain were produced and characterized (*Prox et al., 2013*). These mice were viable and used for prion inoculations performed in this study. First, we analyzed PrP^C levels in A10 cKO and littermate controls at postnatal day (P) 19. Reduction of ADAM10 expression was accompanied by increased PrP^C amounts as revealed by Western blot analysis of cortical homogenates (*Figure 2A*). Residual ADAM10 most likely resulted from glial cells not depleted of ADAM10. In contrast to the cortex, differences in ADAM10 expression and PrP^C levels between A10 cKO and littermate controls were not seen in the cerebellum, a brain region not affected by our knockout strategy (*Casanova et al., 2001*; *Prox et al., 2013*) (*Figure 2A*). Immunohistochemical analysis in P19 mouse brains confirmed the Western blot data by showing increased immunostaining for PrP^C in A10 cKO mice in several regions of the forebrain, as exemplified by the hippocampus and cortex, whereas no such increase was observed in the cerebellum (*Figure 2B*). In addition, a coronal brain section showing costaining of PrP^C with the neuronal marker NeuN correlates with the *Camk2a* driven ADAM10 knockout strategy (*Casanova et al., 2001*) by showing increased PrP^C expression in hippocampal and cortical areas as well as in the striatum (*Figure 2—figure supplement 1*). Increased PrP^C levels in A10 cKO mice did not result from transcriptional upregulation as there were no significant differences in PrP^C mRNA levels in the forebrain at P19 as assessed by quantitative RT-PCR (mean ± SD: controls: 5.96 ± 0.86; A10 cKO: 6.32 ± 1.04; p = 0.23; n = 5) (*Figure 2C*). We also analyzed PrP^C levels in 35-week-old A10 cKO and littermate control mice. In contrast to newborn A10 cKO mice (*Prox et al., 2013*), adult A10 cKO mice did not show differences regarding weight and body size (*Figure 2D* and *Video 1*). Again, as shown at P19, adult A10 cKO mice also showed increased levels of PrP^C in the cortex and hippocampus compared with their littermate controls, as assessed by immunohistochemistry at 35 weeks of age (*Figure 2E*). Taking these observations together, A10 cKO mice are viable and show a temporally unrestricted posttranslational increase of PrP^C in neurons.

Based on previous reports (*Casanova et al., 2001*; *Prox et al., 2013*) and on our findings presented here (*Figure 2*), a qualitative representation showing the regional distribution of the neuronal *Camk2a*-Cre expression (and thus of the ADAM10 knockout and increased PrP^C expression) is provided in *Figure 3A*. In addition, this scheme depicts information on our experimental design

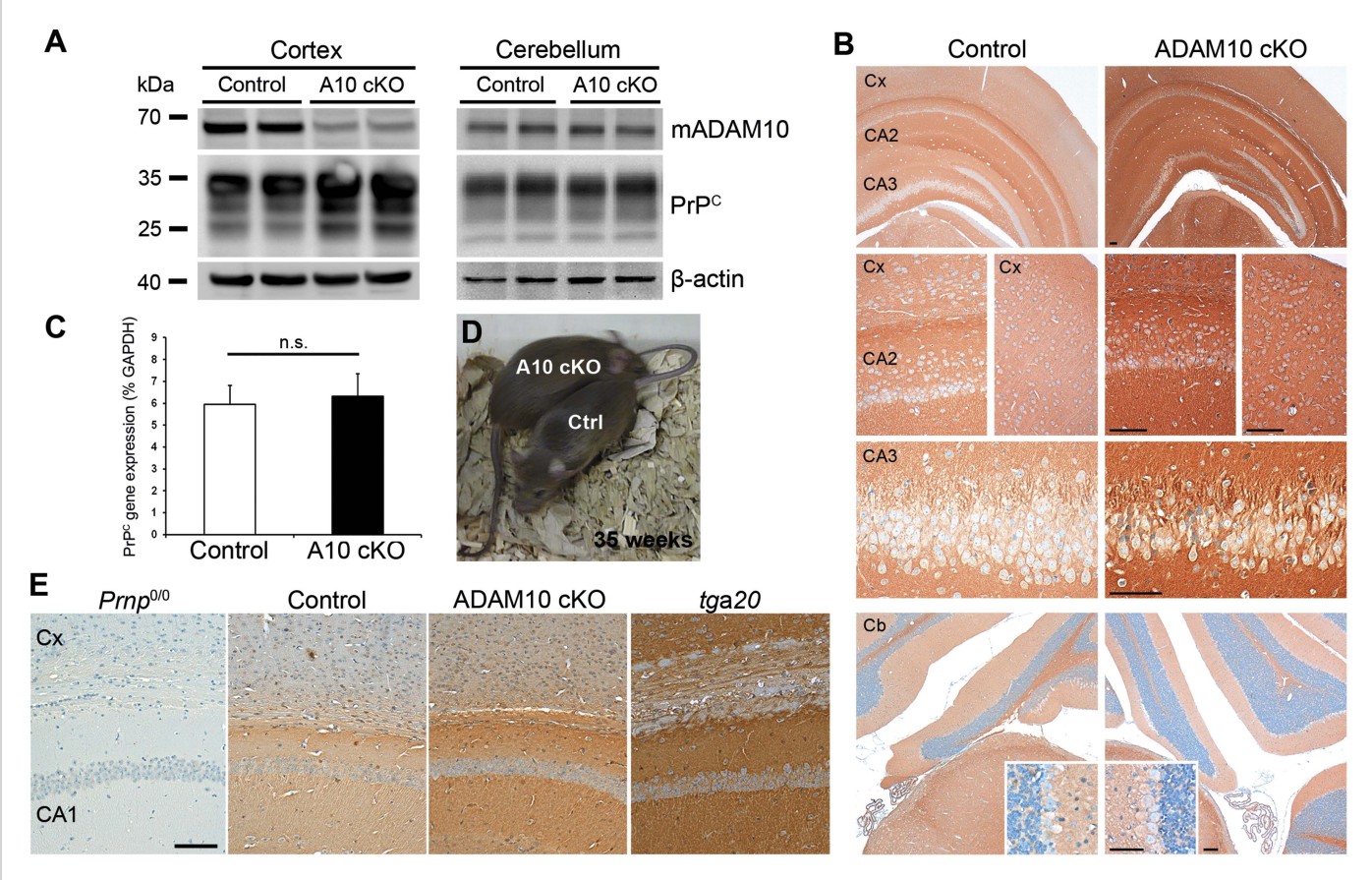

**Figure 2**. Characterization of PrP$^C$ expression in juvenile and adult *Camk2a*-Cre A10 cKO mice and littermate controls. (**A**) Western blot analysis of mature ADAM10 (mADAM10) and PrP$^C$ expression in different brain homogenates from the cortex (left) and cerebellum (right) of A10 cKO and control mice at postnatal day (P) 19. Actin served as a loading control. (**B**) Immunohistochemical detection of PrP$^C$ in forebrain of P19 mice of both genotypes. Overview (top) and magnifications showing cortex (Cx) and hippocampal CA2 and CA3 regions. Overview and details of cerebellum (Cb) are shown below. Scale bars represent 100 µm (insets for Cb: 50 µm). (**C**) Quantitative RT-PCR analysis of PrP$^C$ mRNA levels in A10 cKO mice and controls at P19 (n = 5 for each genotype). GAPDH served as a control for normalization. Error bars indicate SD. (**D**) Adult age-matched A10 cKO and wild-type littermates had a comparable body size at 35 weeks of age. (**E**) Representative immunohistochemical staining of PrP$^C$ in cortex (Cx) and hippocampal CA1 region of adult (35 weeks) A10 cO and control mice. Prion protein knockout (*Prnp*$^{0/0}$) and overexpressing mice (*tga20*) served as negative and positive controls, respectively (scale bar: 100 µm).

The following figure supplement is available for figure 2:

**Figure supplement 1**. Regional distribution of increased PrP$^C$ levels in a coronal brain section of an ADAM10 cKO mouse compared with a wild-type littermate control.

described in detail below, including the site of prion inoculation and brain regions chosen for immunohistochemical or biochemical analyses.

## Drastically reduced incubation time to terminal prion disease in A10 cKO mice

Next, we studied how the absence of the PrP$^C$ sheddase impacts on the course of prion disease. For better orientation, *Figure 3B* summarizes the most important findings presented in the following paragraphs including comparisons of PrP$^{Sc}$ levels, prion-associated neuropathology, and infectivity titers for the different genotypes, brain regions, and time points investigated in this study. Moreover, this overview provides reference to the corresponding figures showing the original data described below.

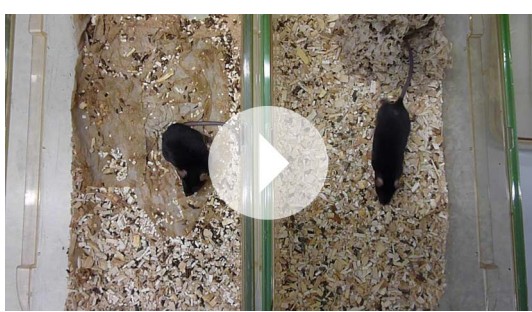

**Video 1.** Clinical presentation of a prion-infected ADAM10 cKO and littermate control mouse at 95 days post inoculation (dpi). Terminally diseased ADAM10 cKO mouse (cage on the left) directly prior to termination of the experiment showing lack of nest-building behaviour and late hypoactivity whereas a wild-type littermate (cage on the right) matched for dpi presents with a regular nest and normal activity.

We inoculated A10 cKO (n = 7) and littermate control mice (n = 8) at 6 weeks of age intracerebrally with the Rocky Mountain laboratory (RML) strain of prions. Mock (CD1)-inoculated A10 cKO mice (n = 5) or wild-type littermates (n = 11) and prion inoculated, prion hypersensitive, PrP^C overexpressing *tga20* mice (n = 8) served as negative and positive controls, respectively. Interestingly, we found significantly reduced incubation times until onset of terminal prion disease in the A10 cKO mice (103 ± 7 days post inoculation (dpi); SD) compared with littermate controls (146 ± 2 dpi) (*Figure 4* and *Table 1*). Clinical manifestations (starting with a neglect of nest formation and fur cleaning, followed by presentation of a stiff tail and gait abnormalities, and terminating with kyphosis, weight loss, and hypoactivity) and disease duration (35 ± 4 days for A10 cKO mice and 40 ± 2.4 days for controls) were comparable between A10 cKO and littermate control mice (*Figure 4*, *Table 1*, *Video 1*). In line with other studies (*Fischer et al., 1996*; *Sandberg et al., 2011*), *tga20* mice showed fastest disease development and had to be sacrificed between 59 and 65 dpi (*Figure 3*) whereas mock-infected A10 cKO and control mice were taken out of the experiment at 200 dpi without any signs of prion disease (*Figure 4* and *Table 1*).

In summary, depletion of the PrP^C sheddase ADAM10 in neurons of the forebrain led to a significant reduction in incubation times of more than 40 days until onset of terminal prion disease without affecting overall clinical presentation.

## Lack of ADAM10 does not influence PrP^Sc loads and neuropathological features of prion disease at terminal disease state

We performed biochemical and neuropathological analyses of terminally prion-diseased A10 cKO and control mice in order to elucidate the reasons for reduced incubation times in the A10 cKO mice. In terminally diseased mice, comparable levels of PrP^Sc were found in frontal brain homogenates of both genotypes as assessed by Western blot analysis after digestion of samples with PK (*Figure 5A*). As expected, no PrP^Sc was found in mock-inoculated controls of each genotype (*Figure 5A* on the right). Since alterations in incubation times may indicate generation of different prion strains (*Barron et al., 2003*) with altered PrP^Sc glycosylation pattern and altered sizes of core fragments of PrP^Sc, we investigated whether lack of ADAM10-mediated shedding resulted in an altered PrP^Sc banding pattern. To this end, we measured the relative proportion of di-, mono-, and unglycosylated forms of PrP^Sc and sizes of PrP^Sc core fragments. No differences in the PrP^Sc glycopattern (*Figure 5B*) or sizes of PrP^Sc core fragments (*Figure 5A*) between terminally diseased A10 cKO and control mice were detected, which argues against the generation of a modified prion strain.

Prion infection caused spongiform lesions, massive astrocytosis, and microgliosis in both genotypes (*Figure 5C* on the left). Similar levels of PrP^Sc immunopositivity were found at this terminal stage of prion disease (*Figure 5C*), thus confirming our biochemical data (*Figure 5A*). Comparable morphological findings were observed in the thalamic brain region (*Figure 5—figure supplement 1*).

Electron microscopic analysis revealed the presence of typical spongiform changes, dystrophic neurites, and abundant autophagy including enlarged multivesicular bodies and autolysosomes in both genotypes (*Figure 5D*). Remarkably, prion-diseased A10 cKO mice presented with an extraordinarily high abundance of tubulovesicular structures (TVS) (*Figure 5—figure supplement 2*) (*Jeffrey and Fraser, 2000*; *Liberski et al., 2010*). These are spherical structures of 25–37 nm diameter that are specific for prion diseases though devoid of PrP (*Jeffrey and Fraser, 2000*; *Liberski et al.,*

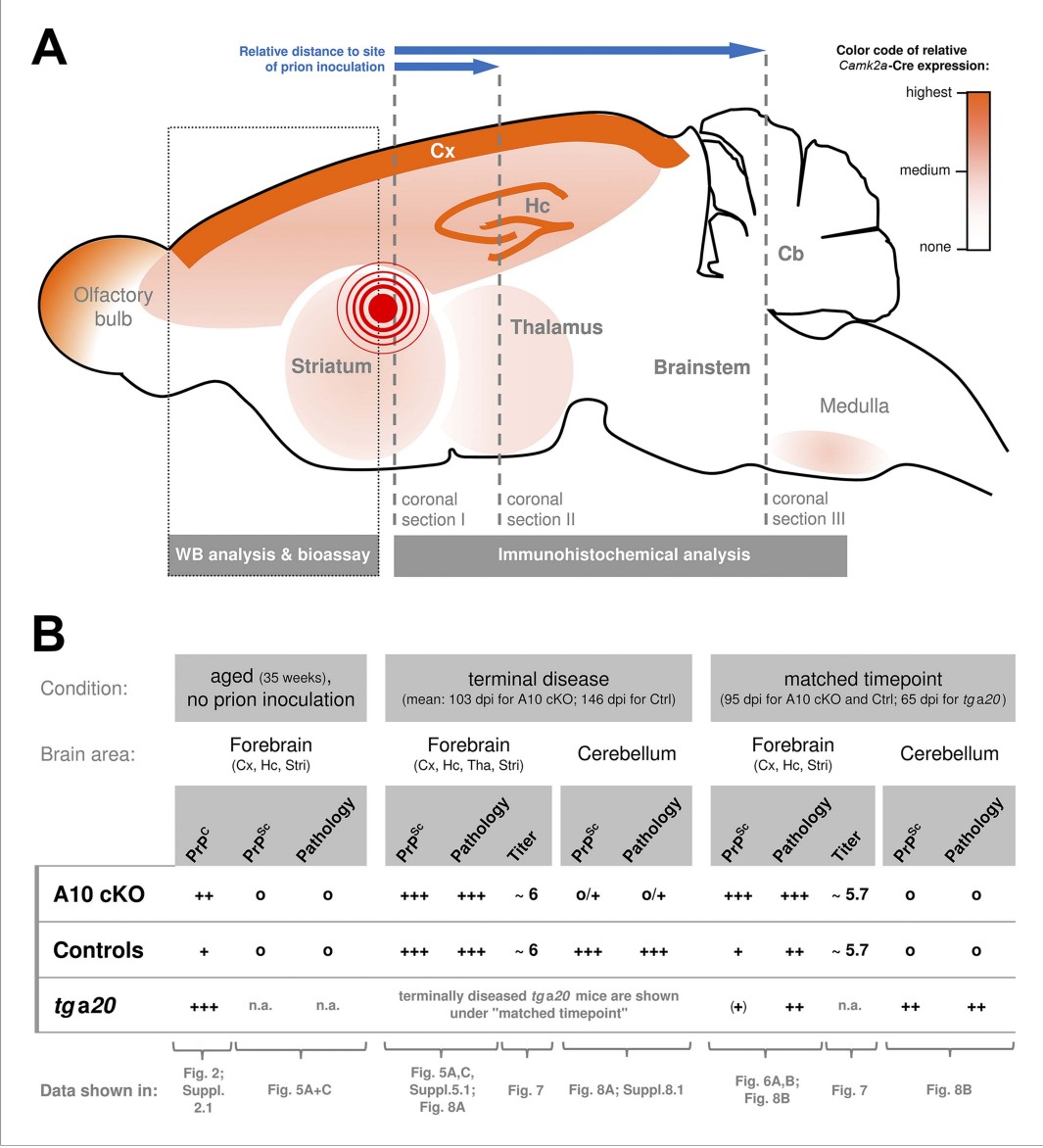

**Figure 3**. Overview of the experimental design and summary of most important findings. (**A**) Scheme of a mouse brain combining a qualitative representation of the *Camk2a* driven ADAM10 knockout strategy and information on the sampling of specimen. The site of intracerebral inoculation of mice with RML prions is indicated by the red encircled dot. Samples of frontal brain (dotted box) were taken for biochemical analysis and determination of infectivity titers (bioassay). The rest of the brain was formalin-fixed and embedded in paraffin. Coronal sections were prepared from different layers (dashed lines) with varying distance to the site of prion inoculation (as indicated by blue arrows) and assessed by immunohistochemical analysis. (**B**) Qualitative comparison of mouse genotypes (A10 cKO, controls, *tga20*) with regard to PrP^C or PrP^Sc levels, prion-associated neuropathology (including spongiosis, astrocytosis, and microglia activation) and prion infectivity titers according to brain region and time point. Reference to corresponding figures showing original data is provided. Cb = Cerebellum; Cx = Cortex; Hc = Hippocampus; Stri = Striatum; Tha = Thalamus; n.a. = not assessed; Qualities: +++ = high/strong; ++ = medium/moderate; + = low/basal; (+) = very low/weak; o = none.

**2008**, **2010**). Thus, our model may allow purification of these structures and could contribute to unraveling the nature and relevance of TVS in prion diseases.

Taken together, these data indicate that, at terminal prion disease, the lack of PrP^C shedding does not influence local PrP^Sc distribution while it does affect the appearance of TVS.

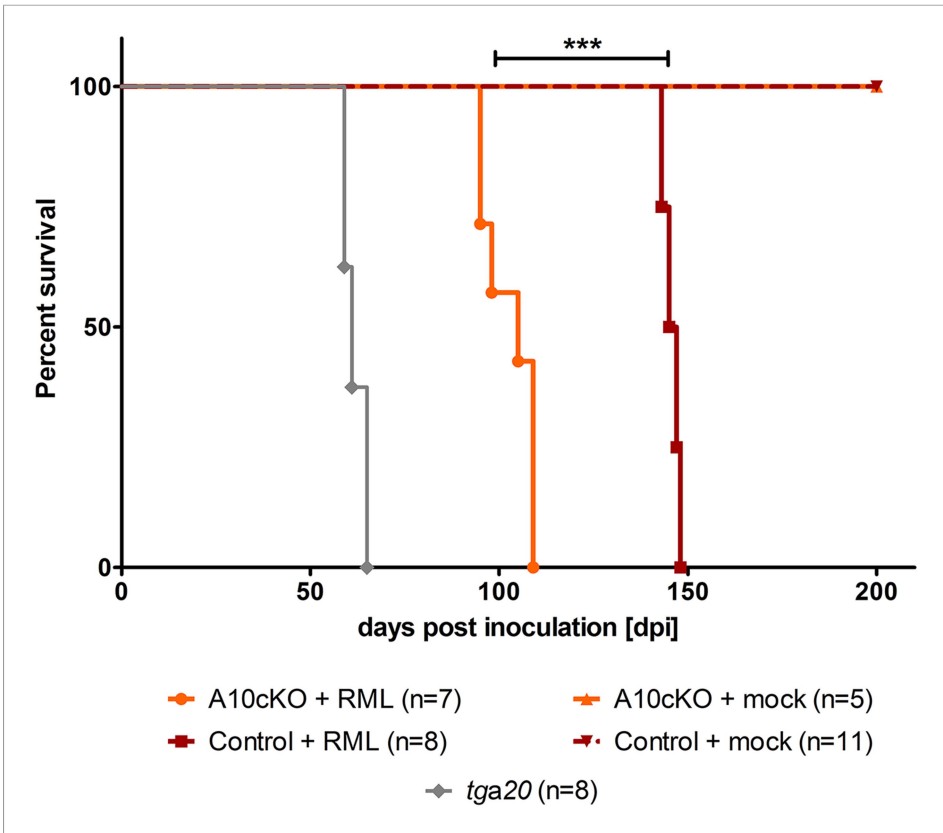

**Figure 4**. Survival curves of mice upon intracerebral inoculation with RML prions. Age-matched juvenile (6-week-old) A10 cKO mice (n = 7), littermate control mice (n = 8), and *tga20* mice (n = 8) were intracerebrally inoculated with RML prions (+RML) and time until development of terminal prion disease was measured as days post inoculation (dpi; ***p = 7.6 × 10⁻⁷). As a negative control (mock), 5 A10 cKO mice and 11 littermate controls were intracerebrally inoculated with brain homogenate of CD1 mice and sacrificed without clinical signs at 200 dpi.

## Enhanced PrP<sup>Sc</sup> formation and calpain levels in A10 cKO mice

To elucidate the order of events underlying changed disease kinetics, we measured PrP$^{Sc}$ production in A10 cKO mice and littermate controls. Since in A10 cKO mice levels of PrP$^{C}$ are posttranslationally increased, we also investigated this issue in *tga20* mice with genetically increased *Prnp* expression. To this aim, we performed Western blot analysis of total PrP (i.e., PrP$^{C}$ plus PrP$^{Sc}$) and PK-resistant PrP$^{Sc}$ in A10 cKO and littermate control mice matched for days post prion inoculation (95 dpi) as well as in terminally prion-diseased *tga20* mice (65 dpi). Relative levels of total PrP were highest in *tga20* mice (5.0 ± 0.3; SEM), with increased levels in A10 cKO mice (2.8 ± 0.4) compared with controls (set to 1 ± 0.1) (A10 cKO vs control: **p = 0.0022; *tga20* vs control: ***p = 0.0005; A10 cKO vs *tga20*: **p = 0.0015) (*Figure 6A*). Surprisingly, relative levels of PrP$^{Sc}$ were significantly elevated in A10 cKO mice (3.9 ± 0.8), with moderate levels in littermate controls (set to 1 ± 0.2) and reduced levels in *tga20* mice (0.6 ± 0.3) (A10 cKO vs control: *p = 0.049; A10 cKO vs *tga20*: *p = 0.032). There were no atypical PrP patterns as assessed by PK digestion at 4°C ('cold PK') or with lower dilutions of PK (*Figure 6—figure supplement 1*). In summary, A10 cKO mice showed significantly increased PrP$^{Sc}$ formation compared with littermate controls and *tga20* mice, which cannot solely be explained by increased neuronal PrP$^{C}$ levels.

Neuropathological changes in the forebrain, including spongiosis, astrocytosis and microgliosis, were more prominent in A10 cKO mice (*Figure 6B*). Remarkably, severe differences could be observed for PrP$^{Sc}$ immunopositivity, with A10 cKO mice showing prominent PrP$^{Sc}$ deposits in cortical and hippocampal regions while controls and *tga20* mice showed only moderate levels, thus confirming the biochemical data (*Figure 6A*).

**Table 1**. Clinical presentation of A10 cKO, littermate control and *tga20* mice inoculated with RML prions

| Clinical manifestations | RML prions | | | CD1 mock | |
|---|---|---|---|---|---|
| | A10 cKO (n = 7) | Controls (n = 8) | tga20 (n = 8) | A10 cKO (n = 5) | Controls (n = 11) |
| First clinical signs (dpi) | 68 (±3.5) | 106 (±2.5) | n.a. | None | None |
| Duration of clinical signs (d) | 35 (±3.8) | 40 (±2.4) | n.a. | - | - |
| Time to terminal disease (dpi) | 103 (±6.6) | 146 (±2.1) | 62 (±2.8) | * | * |
| Progression of signs | Steady | Steady | Rapid | – | – |
| Quitting of nest-building | All | All | All | – | – |
| Ungroomed coat | Most (6/7) | Most (6/8) | Rare (2/8) | – | – |
| Stiff tail | All | Most (7/8) | Most (5/8) | – | – |
| Gait disturbance | All | All | All | – | – |
| Hind leg paresis | Rare (2/7) | Rare (3/8) | Most (7/8) | – | – |
| Kyphosis | Most (6/7) | Most (6/8) | Rare (3/8) | – | – |
| Weight loss | All | All | Most (6/8) | – | – |
| Late hypoactivity | All | All | Most (5/8) | – | – |

Negative controls included A10 cKO and littermate controls inoculated with CD1 brain homogenates (mock). Asterisks indicate that CD1 mock-inoculated animals were sacrificed at 200 days post inoculation (dpi) without any clinical signs.

Next, we addressed the question whether elevated (membrane) levels of PrP$^C$ in A10 cKO and *tga20* mice caused an increase in PrP$^C$-mediated neurotoxic signaling events. Therefore, we biochemically assessed two candidate signaling pathways, via the Src kinase Fyn and the MAP kinases Erk1/2, that have previously been reported to be activated upon binding of toxic oligomers to PrP$^C$ (*Mouillet-Richard et al., 2000*; *Schneider et al., 2003*; *Larson et al., 2012*; *Um et al., 2012*; *Pradines et al., 2013*). However, we did not observe any significant differences in the activation state of these kinases in forebrain homogenates between *tga20*, A10 cKO, and littermate control mice at the chosen time point (*Figure 6C*).

A10 cKO mice showed significantly increased PrP$^{Sc}$ production (*Figure 6A,B*) compared with controls (at 95 dpi) and *tga20* mice (at 65 dpi). Since PrP$^{Sc}$-associated neurotoxicity and cell death have been linked to formation of membrane pores by binding and clustering of PrP$^{Sc}$ to PrP$^C$ at the plasma membrane resulting in calpain upregulation (*Falsig et al., 2012*; *Sonati et al., 2013*), we investigated this in our mice. Specifically, we biochemically analyzed calpain expression in the frontal brain of all experimental groups at the aforementioned time points. Interestingly, we found significant upregulation of calpain expression in A10 cKO mice (1.78 ± 0.24; SEM) compared with littermate controls (set to 1 ± 0.07; *p = 0.034) and *tga20* mice (0.93 ± 0.14; *p = 0.022) (*Figure 6D*), thus correlating with PrP$^{Sc}$ levels (*Figure 6A,B*). This seemed to be specific for prion disease as no differences were detected in non-prion infected A10 cKO mice (mean: 0.90 ± 0.26 SEM; n = 4) when compared with age-matched wild-type controls (set to 1 ± 0.19; n = 5; p = 0.7) (*Figure 6—figure supplement 2A*). In prion disease we observed a reduction of the described calpain substrates spectrin (*Falsig et al., 2012*) and neuron-specific activator p35 (*Lee et al., 2000*), suggesting that increased expression in A10 cKO mice correlated with the activity of calpain (*Figure 6—figure supplement 2B,C*). However, it should be noted that we failed to detect differences in specific cleavage products of spectrin and p35.

Our data suggest that lack of ADAM10-mediated PrP$^C$ shedding leads to increased production of PrP$^{Sc}$ and upregulation of calpain expression, whereas activation of proposed toxic PrP$^C$-dependent signaling pathways was not observed.

## Similar titers of prion infectivity in brain samples of A10 cKO and control mice at terminal and preclinical phases of prion disease

We next investigated whether reduced incubation times and increased PrP$^{Sc}$ conversion rates in A10 cKO mice were accompanied by elevated prion infectivity titers and therefore performed bioassays

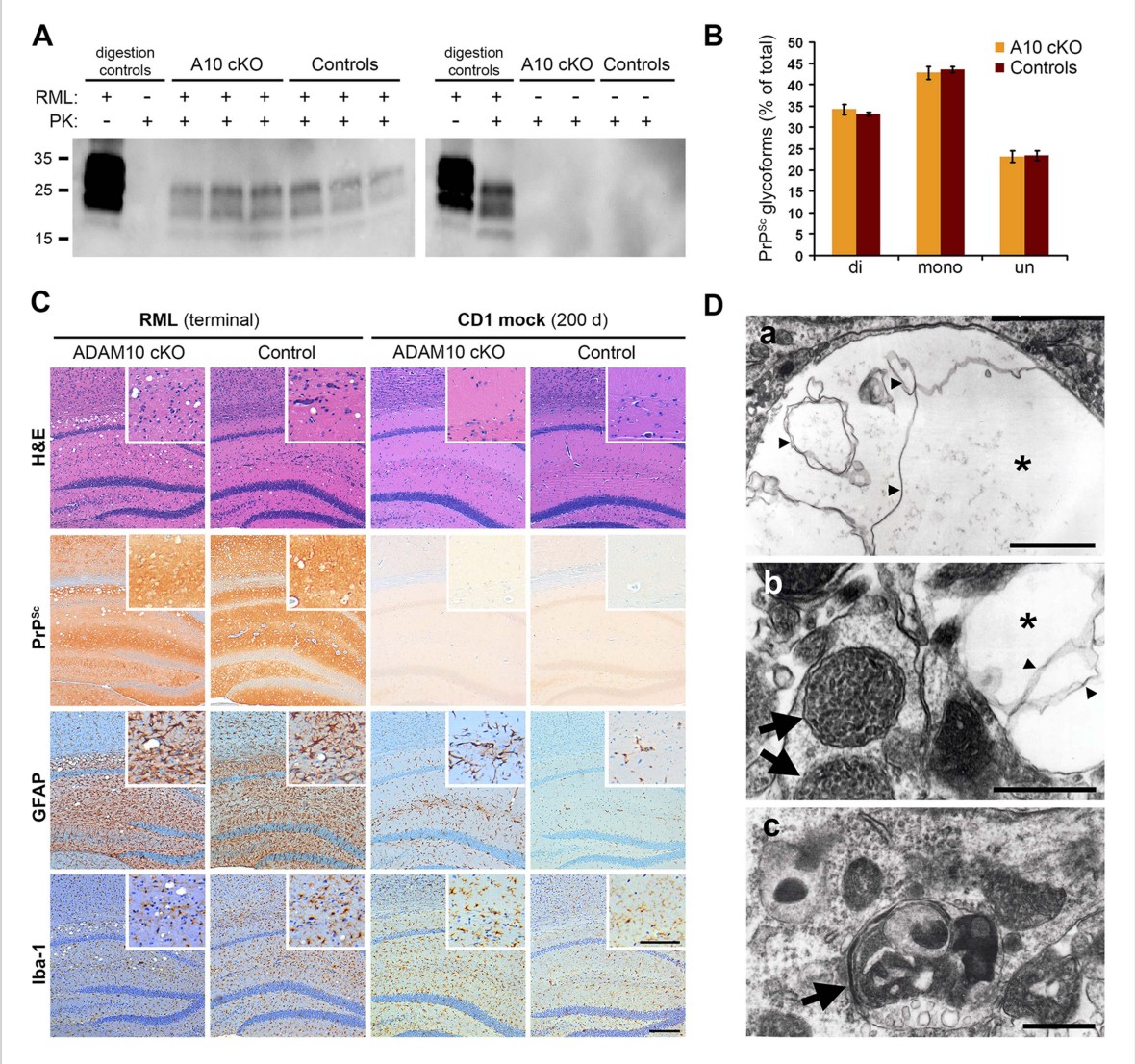

**Figure 5**. Analysis of terminally prion-diseased A10 cKO mice and littermate controls. (**A**) Detection of PrP$^{Sc}$ by Western blot analysis of proteinase K (PK)-digested (PK+) forebrain homogenates of prion-infected (RML+; blot on the left) and mock-inoculated (RML−; blot on the right) A10 cKO mice and controls at terminal stage of disease. Samples from three (RML+) or two (RML−) representative animals per genotype are shown. Digestion controls included prion-diseased/undigested (RML+/PK−), prion-negative/PK-digested (RML−/PK+), or prion-diseased/digested (RML+/PK+) brain homogenates. (**B**) Comparison of proportions of di-, mono, and unglycosylated PrP$^{Sc}$ forms between A10 cKO (n = 4) and control mice (n = 4; error bars indicate SEM). (**C**) Representative histological analysis in the forebrain of terminally prion-diseased (RML terminal) and healthy mock-inoculated (CD1 mock) A10 cKO and control mice including H&E staining and immunostaining of PrP$^{Sc}$, GFAP (for detection of astrocytes), and Iba-1 (for detection of microglia). Scale bar in overviews: 200 µm; scale bar for inlays showing magnifications of representative areas: 100 µm. (**D**) Electron microscopy photographs showing intraneuronal vacuoles (asterisk in (a) and (b)) containing membranous structures (small arrowheads), enlarged and densely packed multivesicular bodies (arrows in (b)) as well as autophagic membranes (arrow in (c)) and autolysosomes that were found in terminally prion-infected forebrains of both genotypes (exemplified here for an A10 cKO brain). Scale bars represent 500 nm.

The following figure supplements are available for figure 5:

**Figure supplement 1**. Neuropathological features in the thalamus of ADAM10 cKO and control mice.

**Figure supplement 2**. Electron microscopic analysis.

(*Figure 7*). We intracerebrally inoculated forebrain homogenates of A10 cKO and littermate control mice at terminal (∼103 dpi for A10 cKO and ∼146 dpi for controls; see *Table 1*) and preclinical (60 dpi, 35 dpi) prion disease into *tga20* (n = 4 for each sample) reporter mice. In addition, we assessed the 95

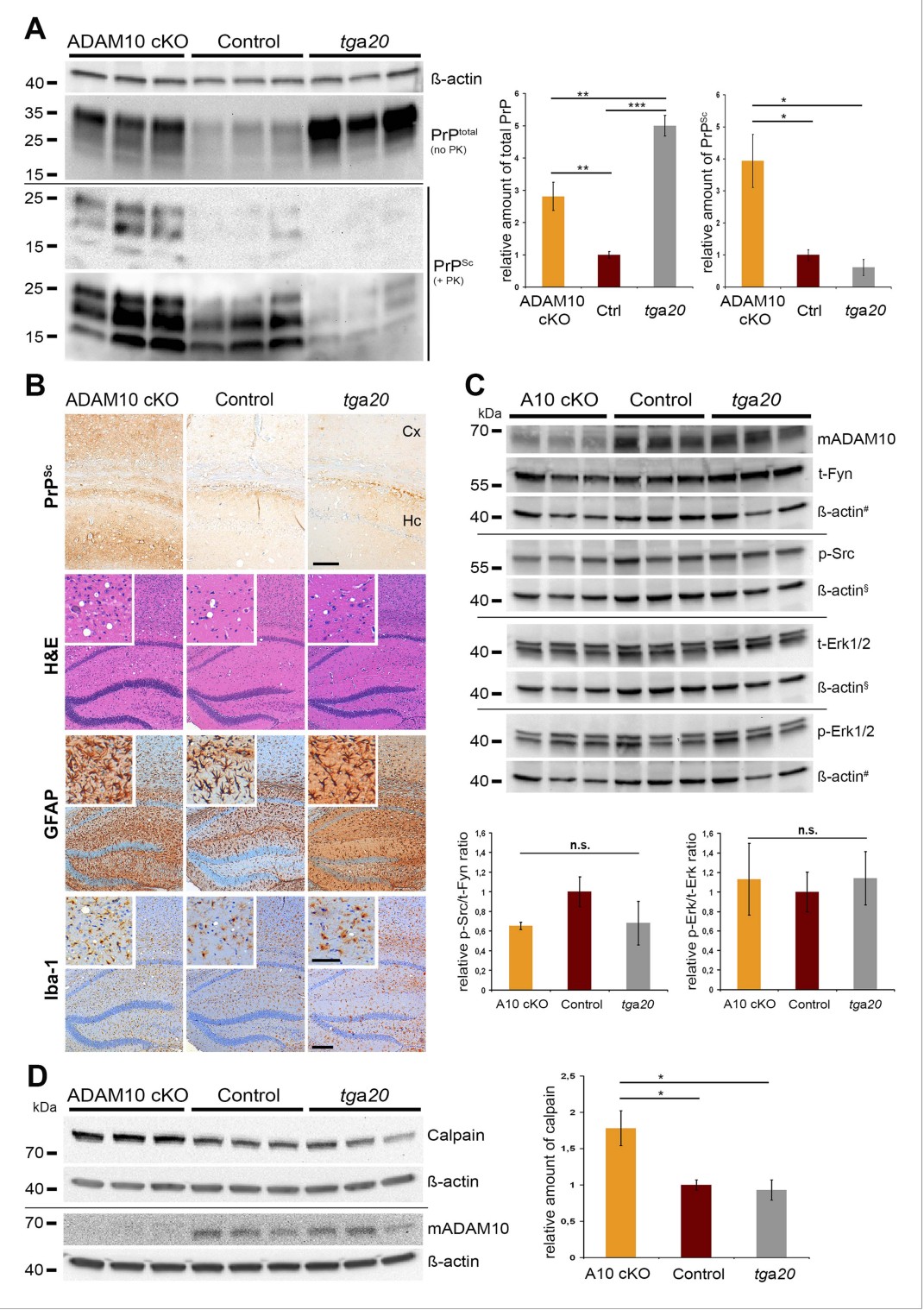

**Figure 6**. PrP^Sc formation, neuropathology, toxic signaling, and calpain levels at a matched time point. (**A**) Assessment of total PrP (no proteinase K (PK)) and PrP^Sc amounts (+PK; blot is shown with short and longer exposition) by parallel replica Western blot analysis in forebrain homogenates of age-matched A10 cKO mice and littermate controls (both at 95 days post inoculation (dpi); n = 3 for each genotype) as well as terminally diseased *tga20* mice (at 65 dpi; n = 3). Actin was detected in the undigested homogenates (no PK) and served as loading control. Densitometric quantification of relative protein amounts from two technical replicates is shown on the right. (**B**) Morphological analysis of neuropathological lesions in forebrains (showing hippocampal and cortical brain

*Figure 6. continued on next page*

*Figure 6. Continued*

regions) of A10 cKO, littermate controls, and *tga20* mice at the aforementioned time points (scale bars: 200 μm in overviews and 100 μm in insets and for PrP$^{Sc}$). (**C**) Biochemical assessment of candidate toxic signaling pathways showing protein levels of total Fyn (t-Fyn), phosphorylated (Tyr416) Src (p-Src) as well as total (t-Erk1/2) and phosporylated (Thr202/Tyr204) Erk1/2 (p-Erk1/2). Actin served as a loading control and for normalization (# and § indicate use of the same actins as corresponding signaling proteins were detected on the same Western blot). Quantitative densitometric analysis of relative p-Src/t-Fyn (left) and p-Erk/t-Erk ratio (right) (n.s. = not significant). (**D**) Representative Western blot analysis (left) and quantification of three technical replicates (right) of calpain levels in aforementioned samples. Levels of ADAM10 are shown in (**C**) and (**D**) to confirm the ADAM10 status. Error bars indicate SEM; *p <0.05; **p <0.01, ***p <0.001 (p values of Student's t-test are given in the main text).

The following figure supplements are available for figure 6:

**Figure supplement 1**. 'Cold proteinase K (PK)' and partial PK digestion of forebrain samples from ADAM10 cKO, control and *tga20* mice at 95 days post inoculation (dpi).

**Figure supplement 2**. Calpain levels and calpain substrates in forebrain homogenates of ADAM10 cKO and control mice.

dpi time point (n = 6 *tga20* mice per sample) at which A10 cKO mice were clinically affected while controls were still asymptomatic and where we observed significant differences in PrP$^{Sc}$ loads (*Figure 6A,B*). Titers of infectious prions increased with progression of disease, showed only some inter-individual variation, and reached a plateau phase towards the terminal stage of disease. However, they were independent of ADAM10 expression since we did not find differences between the two genotypes. At early time points (35 dpi) prion titers were low (mean: A10 cKO: 3.9 ± 0.2 logLD50; control: 4.0 ± 0.2), intermediate titers were seen at 60 dpi (A10 cKO: 5.0 ± 0.5; control: 4.9 ± 0.8), and highest titers were observed in terminally ill mice (A10 cKO: 5.8 ± 0.2; control: 6.0 ± 0.2) (*Figure 7*). Interestingly, significantly increased PrP$^{Sc}$ amounts in A10 cKO mice at 95 dpi (*Figure 6A,B*) did not result in elevated prion infectivity titers (A10 cKO: 5.74 ± 0.05; control: 5.66 ± 0.08; p = 0.2).

## ADAM10-mediated shedding might contribute to spreading of prion pathology

Since PrP$^{Sc}$ is shed from the plasma membrane in vitro (*Taylor et al., 2009*) and anchorless PrP$^{C}$ can convert into PrP$^{Sc}$ (*Chesebro et al., 2005*, *2010*; *Stöhr et al., 2011*), we were interested to investigate if lack of PrP shedding affects the spread of prion pathology within the brain. Therefore, we performed a morphology-based spatiotemporal analysis in prion-infected A10 cKO and littermate control mice.

We assessed a brain region in direct proximity to the site of prion administration (striatum) and brain regions (cerebellum and brain stem) distant from the inoculation site and not directly affected by the depletion of ADAM10 (*Figure 2*; see *Figure 3* for overview) (*Casanova et al., 2001*; *Prox et al., 2013*). Whereas for striatum we observed neuropathological changes typically found in terminal prion disease irrespective of the ADAM10 status, we found an apparently reduced degree of spongiosis in the cerebellum and brain stem of A10 cKO mice compared with terminally diseased littermate controls (*Figure 8A* and *Figure 8—figure supplement 1*). In line with this, other prion-associated pathological hallmarks, including PrP$^{Sc}$ immunopositivity, astrocytosis involving Bergmann glia, and microglia activation, were observed in the cerebellum of controls, but were virtually lacking in this brain region in A10 cKO mice (*Figure 8A*).

Reduced disease duration in A10 cKO mice leading to diminished dissemination of prion-associated pathology may represent a possible explanation for these findings. In order to control for this, we assessed neuropathological alterations at a matched time point (95 dpi) in both A10 cKO mice and littermate controls. Additionally, we included terminally prion-diseased *tga20* mice (65 dpi). In the striatum, typical neuropathological changes were seen in all groups of mice. Similar to hippocampus and cortex (*Figure 6B*), increased PrP$^{Sc}$ immunopositivity and astrogliosis were seen in A10 cKO mice

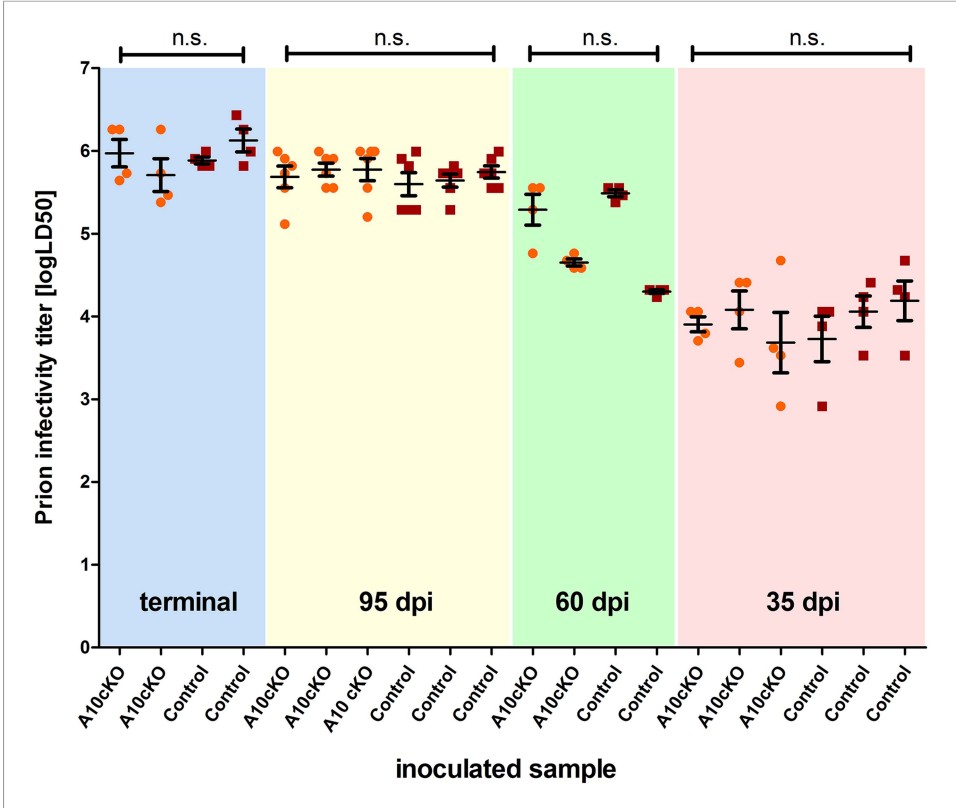

**Figure 7**. Titers of prion infectivity in terminally diseased and preclinical A10 cKO mice and controls. Prion titers in forebrain homogenates of A10 cKO mice and littermate controls at terminal or matched (preclinical) time points (95 days post inoculation [dpi], 60 dpi, and 35 dpi) after prion inoculation as assessed by bioassays in *tga20* reporter mice. Each dot indicates the titer (shown as logLD50) assessed in a single reporter mice. Bars indicate mean values from six (for the 95 dpi time point) or four (all other time points) *tga20* mice. Error bars indicate SD.

(*Figure 8B*). In line with our biochemical findings (*Figure 6A*), *tga20* mice showed lowest PrP$^{Sc}$ amounts in the forebrain. Interestingly, in the cerebellum and brain stem, relevant prion-typical neuropathological changes were observed in *tga20* mice at 65 dpi but were almost absent in both A10 cKO and littermate controls at 95 dpi (*Figure 8B*). This argues against a temporal cause for the observed differences and rather supports an accelerating role of shedding in the spread of prion disease within the CNS. Since *tga20* mice show enhanced rates of PrP$^C$ shedding (*Figure 1E*) (*Altmeppen et al., 2011*), our spatiotemporal data may indicate a correlation between PrP$^C$ shedding and the efficiency of prion spread within the brain, with *tga20* mice having the highest efficiency whereas A10 cKO mice show the least efficient dissemination of prion pathology (see model in *Figure 9*).

## Discussion

Proteolytic processing of the prion protein is critically involved in controlling its physiological functions. For α-cleavage, a protective role for both prion and AD is postulated and attributed to destruction of the neurotoxic domain and production of the neuroprotective N1 fragment respectively (*Guillot-Sestier et al., 2009*; *Resenberger et al., 2011*; *Westergard et al., 2011*; *Beland et al., 2012*; *Guillot-Sestier et al., 2012*; *Fluharty et al., 2013*). Moreover, the C1 fragment of PrP$^C$ is involved in myelination (*Bremer et al., 2010*). The protective role of α-cleavage is also supported by the fact that α-cleavage-resistant PrP$^C$ deletion mutants are intrinsically neurotoxic upon expression in mice (*Shmerling et al., 1998*; *Baumann et al., 2007*) (reviewed in *Yusa et al., 2012*). In contrast, our knowledge of the consequences of PrP$^C$ shedding, though being evolutionary conserved and constitutively occurring under physiological conditions, is limited. In a previous study we have shown

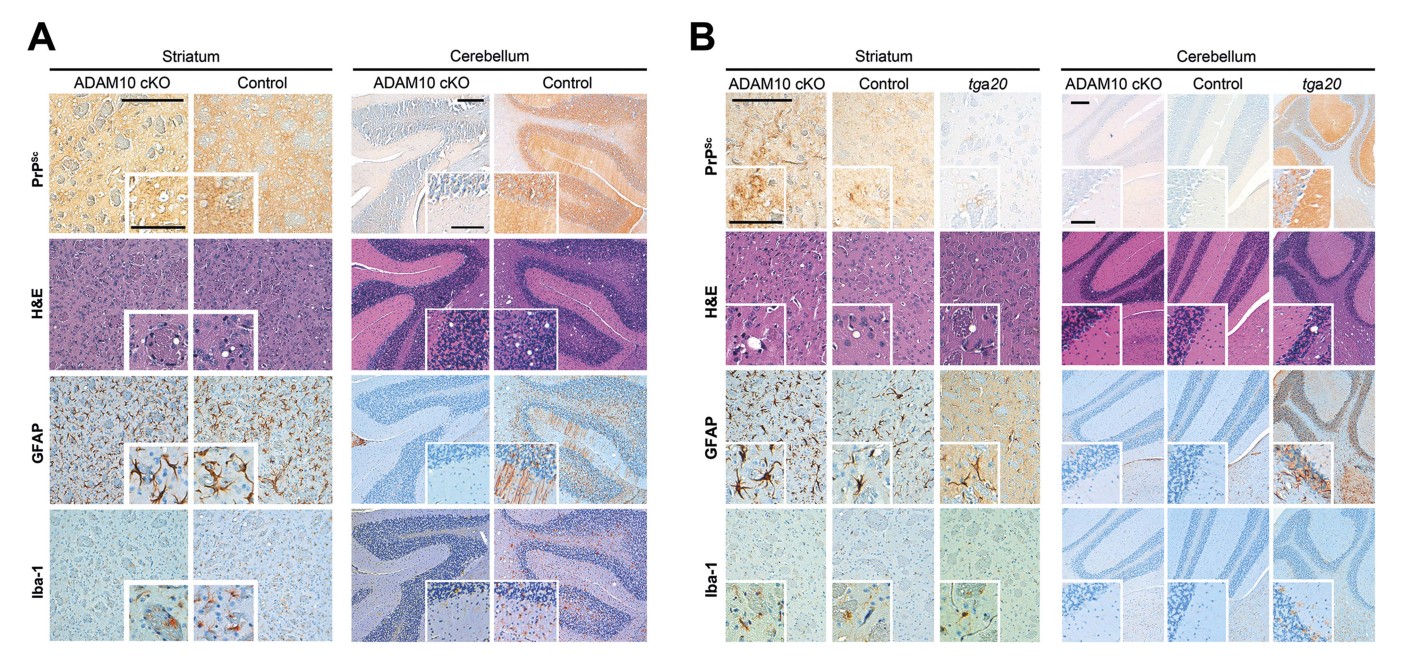

**Figure 8**. Spatiotemporal analysis of prion-associated pathology. (**A**) Comparison of neuropathological features including PrP$^{Sc}$ deposition, spongiotic changes (presented in H&E stainings), astrocyte (GFAP) and microglia (Iba-1) activation in striatum (as a site in close proximity to prion inoculation) and cerebellum (resembling a brain area distant to prion inoculation) in A10 cKO and littermate control mice at a terminal stage of prion disease (i.e., ~103 days post inoculation [dpi] for A10 cKO and ~146 dpi for controls). While similar pathological alterations were observed in the striatum of both genotypes, prion-associated lesions in the cerebellum were almost absent in A10 cKO mice. (**B**) At a matched time point (95 dpi for A10 cKO and littermate controls; 65 dpi [i.e., terminal disease] for *tga20*), prion-related pathology was found in the striatum of all genotypes analyzed. As described earlier for the cortex and hippocampus (*Figure 6*), differences in PrP$^{Sc}$ amounts could also be observed in the striatum. In the cerebellum of both time-matched A10 cKO mice and littermate controls, prion-associated lesions were largely absent whereas *tga20* mice showed all relevant neuropathological features already at 65 dpi. Representative pictures for at least three animals per genotype and time point are shown. Scale bars represent 200 μm (overviews) or 100 μm (insets).

The following figure supplement is available for figure 8:

**Figure supplement 1**. Spongiosis in brain stem and cerebellum of ADAM10 cKO and control mice.

that ADAM10 represents the principal PrP$^C$ sheddase in vivo with lack of ADAM10 resulting in a posttranslational increase of PrP$^C$ levels (*Altmeppen et al., 2011*). This pointed to a key role for this protease in the regulation of PrP$^C$ levels at the neuronal surface, yet its impact on prion diseases remained unknown. We recently generated A10 cKO mice, a viable model with selective postnatal ablation of ADAM10 in neurons of the forebrain (*Prox et al., 2013*). In the present study we demonstrate that prion-inoculated A10 cKO mice show considerably reduced incubation times and increased conversion of PrP$^C$ to PrP$^{Sc}$. In contrast, spread of prion pathology within the brain is reduced, thus indicating a dual role for ADAM10 in the pathophysiology of prion disease (*Figure 9*).

ADAM10 acts as a sheddase for a number of neuronally expressed proteins with important functions in the development and homeostasis of the brain (*Yang et al., 2006*; *Weber and Saftig, 2012*). For PrP$^C$ it is now widely accepted that ADAM10 is the PrP$^C$ sheddase (*Parkin et al., 2004*; *Taylor et al., 2009*; *Altmeppen et al., 2011*; *Wik et al., 2012*; *Ostapchenko et al., 2013*; *McDonald et al., 2014*). Here we extend our previous findings on increased levels of PrP$^C$ in the absence of ADAM10 (*Altmeppen et al., 2011*) in different cellular models and observe colocalization between PrP$^C$ and ADAM10 at the plasma membrane, which is regarded as the primary site of ADAM10-mediated shedding events (*Lammich et al., 1999*; *Chen et al., 2014*). In addition to accumulation of PrP$^C$ in the early secretory pathway (*Altmeppen et al., 2011*), we here show that lack of ADAM10 also results in elevated membrane levels of PrP$^C$. This supports a model where ADAM10-mediated shedding controls PrP$^C$ membrane homeostasis (*Altmeppen et al., 2013*).

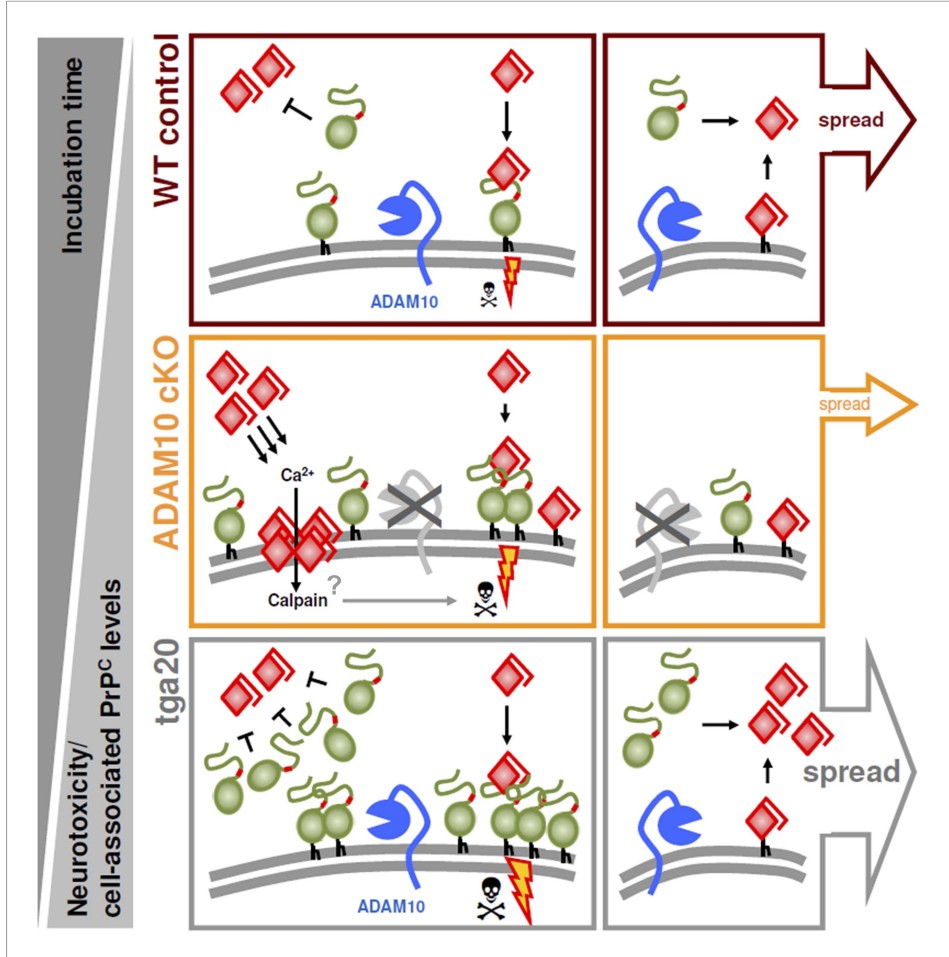

**Figure 9**. Model of the dual role of ADAM10-mediated shedding in prion disease. ADAM10 regulates PrP$^C$ levels at the plasma membrane and releases almost full length PrP into the extracellular space. Thereby it affects (i) neurotoxicity, (ii) PrP$^{Sc}$ formation, and (iii) spreading of prion pathology. (i) Lack of ADAM10 (as assessed here by use of A10 cKO mice) results in elevated PrP$^C$ membrane levels. Membrane levels of PrP$^C$ (as a receptor) likely determine PrP$^{Sc}$-associated neurotoxicity (as indicated by sizes of thunderbolts and skulls) and thereby incubation times with shortest survival in *tga20* mice and reduced incubation times in A10 cKO mice compared with wild-type littermates with longest survival (order reflected by grey triangles on the left). (ii) Shed PrP, which is most efficiently produced in *tga20* and absent in A10 cKO mice, might block formation of PrP$^{Sc}$. This is reflected by the different PrP$^{Sc}$ amounts found in our different experimental groups (A10 cKO > wild-type littermates > *tga20*). The combination of increased PrP$^C$ membrane levels and PrP$^{Sc}$ formation in A10 cKO mice might favor increased production of membrane pores (as indicated in the middle row on the left) and neurotoxic Ca$^{2+}$ influx with possible ('?') involvement of calpain. (iii) Finally, spread of prion-associated pathology within the brain also seems to be affected by the levels of ADAM10 expression since *tga20* mice showed enhanced whereas A10 cKO mice showed reduced dissemination of neuropathological features (as indicated by size of arrowheads on the right). Key references supporting this model are given in the text. Based on this model, stimulation of ADAM10 might therefore offer a treatment option. With regard to incubation times, protective effects by reducing local membrane bound PrP$^C$ amounts and by producing a protective soluble fragment able to block PrP$^{Sc}$ formation seem to predominate the disadvantage of increased spread by production of anchorless prions.

However, cell culture models as well as mice with pan-neuronal depletion of ADAM10 did not allow investigation of the role of PrP shedding in prion diseases (*Altmeppen et al., 2011*). Therefore, we used A10 cKO mice (*Prox et al., 2013*) in which we observed increased PrP$^C$ levels that were not caused by elevated PrP$^C$ mRNA levels.

Onset of terminal prion disease in mice upon intracerebral inoculation with defined prion strains occurs after defined incubation times. Once clinical symptoms appear, individual mice reliably succumb to disease within days. In contrast to peripheral prion inoculation where the integrity of the lymphoreticular or autonomous nervous system plays a decisive role in neuroinvasion of prions, incubation time after intracerebral inoculation is mainly influenced by PrP$^C$ levels (*Mabbott et al., 2000*; *Glatzel et al., 2001*; *Prinz et al., 2003*). This is nicely exemplified by *tga20* mice constitutively overexpressing PrP$^C$ (*Fischer et al., 1996*; *Sandberg et al., 2011*). Our intracerebrally inoculated A10 cKO mice developed prion disease >40 days earlier than controls, corresponding to a shortening of incubation times of ~30%. Such dramatic reductions upon intracerebral inoculation with high-dose RML prions have so far only been achieved when PrP$^C$ is massively overexpressed (*Fischer et al., 1996*). Less dramatic reductions were, for instance, described in mice lacking superoxide dismutase 1 (*Akhtar et al., 2013*) or overexpressing the heat shock protein Hspa13 (*Grizenkova et al., 2012*). The only other study investigating the role of ADAM10 in prion disease showed that transgenic overexpression of bovine ADAM10 in mice led to prolongation of prion disease (*Endres et al., 2009*), a finding which is complementary to our data on the effect of ADAM10 deficiency. However, these authors linked this effect to transcriptional downregulation of PrP$^C$ and not to proteolytic processing events. In contrast, data from the present and previous studies show that transcriptional regulation of PrP$^C$ is not affected by ADAM10 whereas proteolytic processing clearly is (*Parkin et al., 2004*; *Taylor et al., 2009*; *Altmeppen et al., 2011*; *Wik et al., 2012*; *Ostapchenko et al., 2013*; *McDonald et al., 2014*). We considered the possibility that non-PrP dependent effects resulting from ADAM10 deficiency may have contributed to disease acceleration. Indeed, ADAM10 is involved in a number of developmental and immunological processes and A10 cKO mice show learning deficits (*Weber and Saftig, 2012*; *Prox et al., 2013*). It is virtually impossible to dissect out the individual contribution of the multitude of ADAM10 targets to the pathophysiology of prion disease. For instance, microglia contribute to prion pathophysiology and express CD40L, a known ADAM10 substrate. Prion-inoculated CD40L-deficient mice show decreased incubation times (*Burwinkel et al., 2004*). As CD40L is not expressed on neurons and its receptor (CD40) is not processed by ADAM10, we do not expect that this dyad is altered by our neuron-specific knockout strategy (*Schönbeck and Libby, 2001*; *Burwinkel et al., 2004*). Another microglia-related mechanism relevant to prion disease and influenced by ADAM10 is the Cx3cl1/Cx3cr1 signaling complex. The receptor (Cx3cr1) is expressed on microglia whereas its ligand Cx3cl1 (Fractalkine) is an ADAM10 substrate expressed on neurons. Cx3cl1-deficient mice inoculated with prions show slightly decreased incubation times to disease (*Grizenkova et al., 2014*). Therefore, we do not assume that these events explain the strong effect observed in our study. Most other immune system-mediated effects on prion pathogenesis are only relevant for peripheral prion inoculation and neuroinvasion and do not play major roles in intracerebral prion administration (*Aguzzi et al., 2003*). Furthermore, A10 cKO mice do not show any gross morphological abnormalities in the CNS, have a normal life span once they have survived a critical weaning period, and mock inoculated mice behave normally (*Prox et al., 2013*). Finally, we consider that increased PrP$^{Sc}$ levels found in A10 cKO mice strongly argue in favor of PrP-dependent effects of ADAM10 depletion.

Analysis of A10 cKO mice and littermate controls at 95 dpi as well as *tga20* mice at 65 dpi (which represents terminal disease for these mice) revealed a significant difference in PrP$^{Sc}$ formation between A10 cKO and the other two experimental groups. This cannot be explained by the increased presence of PrP$^C$, since PrP$^C$ overexpressing *tga20* mice showed low levels of PrP$^{Sc}$ even at the terminal stage of the disease. How can this be explained? The molecular basis underlying neurodegeneration in prion diseases is under debate and different mechanisms of neurotoxicity have been proposed. In the receptor model, PrP$^C$ binds oligomeric β-sheet rich protein species, such as PrP$^{Sc}$, and mediates neurotoxic signaling (*Mouillet-Richard et al., 2000*; *Lauren et al., 2009*; *Resenberger et al., 2011*; *Larson et al., 2012*; *Um et al., 2012*). In the pore formation model, PrP$^{Sc}$ oligomers, either directly or upon binding to PrP$^C$, form a membrane channel leading to exaggerated Ca$^{2+}$ influx and calpain activation (*Falsig et al., 2012*; *Sonati et al., 2013*). In our study we found evidence for the link between excessive generation of PrP$^{Sc}$ and increased calpain levels in A10 cKO mice, whereas basal (in littermate controls) and low (in *tga20* mice) PrP$^{Sc}$ levels were not associated with calpain upregulation.

*Tga20* mice are hypersensitive to prion disease yet do not show significant levels of PrP$^{Sc}$ even though PrP$^C$ is strongly overexpressed. In a previous study we have shown and re-evaluated here that

ADAM10-mediated shedding of PrP$^C$ in neurons of *tga20* mice is increased approximately threefold (*Altmeppen et al., 2011*). Of note, soluble GPI-anchorless forms of PrP$^C$, which may be regarded as correlates of shed PrP$^C$, and experimentally-induced release of PrP$^C$ in vitro impair PrP$^{Sc}$ formation (*Marella et al., 2002*; *Meier et al., 2003*; *Kim et al., 2009*; *Yuan et al., 2013*). Thus, high levels of shed PrP$^C$ in *tga20* mice may impair PrP$^{Sc}$ formation whereas lack of shed PrP$^C$ in A10 cKO mice may favor PrP$^{Sc}$ formation (*Figure 9*). In line with this concept, PrP$^{Sc}$ production was decreased in mice moderately overexpressing ADAM10 (*Endres et al., 2009*). In both *tga20* and A10 cKO mice, increased levels of PrP$^C$ at the plasma membrane should facilitate PrP$^C$-mediated neurotoxicity and thereby determine incubation times. Interestingly, these data provide a potential explanation for the *tga20* paradox of low PrP$^{Sc}$ formation despite high PrP$^C$ levels and short incubation times (*Fischer et al., 1996*).

A10 cKO mice and controls showed identical prion titers. On the other hand levels of PrP$^{Sc}$ were elevated and no atypical protease-sensitive forms of PrP$^{Sc}$ were found. It is becoming increasingly evident that titers of prion infectivity, PrP$^{Sc}$, and the presence of potentially neurotoxic PrP conformers are not congruent (*Lasmezas et al., 1997*; *Barron et al., 2007*; *Piccardo et al., 2007*; *Krasemann et al., 2013*). In fact, the exact composition of infectious 'prions' (i.e., the entity determined by bioassay) is not fully understood and it may be that ADAM10-mediated shedding affects formation of PrP$^{Sc}$ while it does not influence production of 'prions'.

We did not observe induction of proposed neurotoxic signaling pathways (*Mouillet-Richard et al., 2000*; *Schneider et al., 2003*; *Larson et al., 2012*; *Um et al., 2012*; *Pradines et al., 2013*). It may be that these pathways occur spatially and temporally restricted and future studies will have to address this question using refined methods of analysis. Alternatively, other yet to be discovered pathways may be involved. In A10 cKO mice, the combination of elevated PrP$^C$ membrane levels and increased PrP$^{Sc}$ amounts may favor increased pore formation and Ca$^{2+}$ influx as indicated by elevated calpain levels. However, since we did not detect unequivocal evidence for calpain activation, further studies are required to substantiate this model.

In view of the fact that prion-infected mice expressing anchorless PrP$^C$ show PrP$^{Sc}$ formation and deposition in ectopic sites (*Lee et al., 2011*) and prion propagation in the CNS (*Chesebro et al., 2005*), we investigated whether shedding contributes to the spread of prion pathology. In fact, A10 cKO mice show prion-related pathology and PrP$^{Sc}$ deposition at the site of (striatum) and adjacent to prion administration (hippocampus, frontal cortex, thalamus), whereas brain regions distant from the inoculation site (cerebellum, brain stem) were unaffected, even at terminal stages of disease. In contrast, in *tga20* mice, dissemination of prion-related pathology occurred efficiently at a much earlier time point. Non-PrP-dependent effects resulting from ADAM10 deficiency, such as altered microglial activity or reduced processing of proinflammatory molecules, may have contributed to impaired spreading (*Weber et al., 2013*; *Prox et al., 2013*). However, the influence of the immune system, including microglial activation on intracerebral spread of prions, is limited, therefore a correlation between PrP shedding and the efficiency of prion spread is possible (*Glatzel et al., 2004*; *Grizenkova et al., 2014*). In line with our findings, it has been shown that, upon prion infection of mice, heterozygous expression of normal and anchorless PrP results in earlier death compared with wild-type mice (*Chesebro et al., 2005*). Thus, shedding of either PrP$^C$ (followed by cell-free conversion [*Kocisko et al., 1994*]) or PrP$^{Sc}$, both of which have been shown to occur (*Taylor et al., 2009*; *Altmeppen et al., 2011*), should be considered as an alternative mechanism of prion spread (*Figure 9*) aside from direct transsynaptic cell-to-cell transfer (*Glatzel et al., 2001*; *Shearin and Bessen, 2014*), exosomal (*Alais et al., 2008*) and viral transfer (*Leblanc et al., 2006*), or tunneling nanotubes (*Gousset et al., 2009*). It remains to be investigated whether ADAM10-mediated shedding is also involved in the release of bona fide prions into body fluids such as cerebrospinal fluid, blood, or nasal secretions, which may potentially increase the risk of transmission (*Tagliavini et al., 1992*; *Perini et al., 1996*; *Parizek et al., 2001*; *Bessen et al., 2010*).

In conclusion, our data indicate that proteolytic processing, as described here for the shedding of the prion protein, represents a master switch in the pathophysiology of prion diseases by drastically affecting PrP$^{Sc}$ formation and incubation times. Interestingly, protective effects of ADAM10 might be hijacked in prion diseases since a reduction of the protease in prion disease has recently been reported in vitro and in vivo (*Chen et al., 2014*). By contrast, shedding also seems to affect the spread of prion pathology within the CNS. Understanding this dual role of ADAM10 in prion disease brings together current concepts of prion biology and might reveal a mechanistic insight into important

pathophysiological processes. Apart from prion disease, in other neurodegenerative proteinopathies, where PrP$^C$-PrP$^{Sc}$ conversion does not play a role, ADAM10-mediated shedding of PrP$^C$ might solely be protective due to reduction of the receptor as well as the production of a soluble blocker of toxic oligomers. In view of a potential role of PrP$^C$ in AD, further research on proteolysis as a regulatory event in prion biology will likely impact on more common neurodegenerative conditions.

## Materials and methods

### Description of mice and ethics statement

In this study we took advantage of the recently generated *Camk2a*-Cre *Adam10* conditional knockout (A10 cKO) mouse model with a depletion of *Adam10* in neurons of the forebrain. Generation as well as a detailed description of these mice can be found elsewhere (*Prox et al., 2013*). Assessment of floxed alleles (*Jorissen et al., 2010*) and the *Camk2a*-Cre status (*Casanova et al., 2001*; *Prox et al., 2013*) in transgenic mice was performed by PCR. Our breeding strategy of crossing *Adam10*$^{F/F}$ with *Camk2a*-Cre:*Adam10*$^{F/+}$ yielded approximately 15% viable Cre-positive *Adam10*$^{F/F}$ offsprings (A10 cKO) (*Prox et al., 2013*). Cre-negative *Adam10*$^{F/F}$ or *Adam10*$^{F/+}$ littermates were used as wild-type controls. In addition, we used prion protein overexpressing *Tg(Prnp)a20* mice (herein referred to as *tga20*) (*Fischer et al., 1996*) as controls for our biochemical and immunohistochemical analysis and as reporter mice in our bioassays for determination of prion infectivity titers as described below. Finally, brain sections of prion protein knockout (*Prnp*$^{0/0}$) mice (*Büeler et al., 1992*) were used as an internal negative control for our immunohistochemical staining procedure.

Our study was carried out in accordance with the principles of laboratory animal care (NIH publication No. 86-23, revised 1985) as well as the recommendations in the Guide for the Care and Use of Laboratory Animals of the German Animal Welfare Act on protection of animals. The protocol was approved by the Committee on Ethics of the *Freie und Hansestadt Hamburg—Amt für Gesundheit und Verbraucherschutz* (permit number 48/09, 81/07 and 84/13).

### Prion inoculations

Inoculations of mice were performed under deep ketamine and xylazine hydrochloride anesthesia. All efforts were made to minimize suffering of the animals including careful observation and special treatment (i.e., administration of wet food and incubation on a warming plate) of mice immediately after inoculations until recovery. In brief, 6–9-week-old A10 cKO mice (n = 12), littermate controls (n = 17), and *tga20* mice (n = 11) were inoculated with 30 µl of 1% homogenate of Rocky Mountain Laboratory (RML) prions (RML 5.0 inoculum, corresponding to $3 \times 10^5$ LD50) into the forebrain. Mice were checked on a weekly basis and observation was intensified to a two-day schedule following the appearance of first clinical signs. To assess disease characteristics in preclinical mice, A10 cKO, wild-type littermates, and *tga20* mice (n = 3 for each genotype) were sacrificed at 35 dpi. Moreover, two A10 cKO mice and two wild-type control mice were taken at 60 days for determination of preclinical prion titers. For matched comparison with terminal A10 cKO mice, preclinical littermate controls (n = 4) were sacrificed at day 95 after inoculation. All other mice (A10 cKO: n = 7; controls: n = 8; *tga20*: n = 8) were sacrificed and analyzed when they reached terminal prion disease (*Table 1*). Additionally, we performed mock inoculations with 30 µl of 1% brain homogenate from uninfected CD-1 mice into age-matched A10 cKO (n = 5) and littermate controls (n = 11). These animals were sacrificed at 200 dpi lacking any signs of prion disease (see *Table 1*).

For the assessment of titers of prion infectivity in our preclinical and terminally diseased A10 cKO and control mice, we performed bioassays in *tga20* reporter mice. To this end, 20 µl of a 1% homogenate of frontal brain derived from an animal to be investigated were intracerebrally inoculated into four or six *tga20* mice. Prion titers (*y*; shown as logLD50) in the original sample were then calculated according to the equation $y = 11.45 - 0.088 \times x$, with *x* being the time to terminal disease of *tga20* reporter mice (in dpi).

### Primary neurons, cell culture, and analysis of media supernatants

Generation of primary neurons of E14 embryos of $^{Nestin}$A10 KO, wild-type C57BL/6, *Prnp*$^{0/0}$ and *tga20* mice as well as immunoprecipitation of shed PrP$^C$ from conditioned media was described (*Altmeppen et al., 2011*).

MEFs of A10 KO and control mice were initially described in (*Hartmann et al., 2002*) and maintained under standard cell culture conditions. Conditioned media were collected after 18–24 hr and prepared for immunoprecipitation of PrP$^C$ (*Altmeppen et al., 2011*). Alternatively, media supernatants were concentrated 10× using ultrafiltration spin columns (Vivaspin 500, Sartorius Stedim Biotech, Goettingen, Germany).

## Western blot analysis

Samples of frontal brain were processed as 10% (wt/vol) homogenates in RIPA buffer (50 mM Tris–HCl pH 8, 150 mM NaCl, 1% NP40, 0.5% Na-Deoxycholat, 0.1% SDS) supplemented with Complete Mini protease inhibitor cocktail (PI; Roche Diagnostics, Mannheim, Germany). Samples were smashed 30× on ice using a Dounce homogenizer and subsequently incubated on ice for 15 min before resuspending by 15× pipetting up and down. Homogenates were centrifuged at 12,000×*g* for 6 min at 4°C and total protein content in supernatants was determined by colorimetric analysis (QuickStart Bradford 1× Dye, Biorad, Hercules, CA) following the manufacturer's instructions. Samples were then normalized to yield equal protein amounts and boiled in 4× loading buffer (250 mM Tris–HCl, 8% SDS, 40% glycerol, 20% β-mercaptoethanol, 0.008% Bromophenol Blue, pH 6.8) for 6 min at 96°C. Gel electrophoresis was performed using 20–50 μg of total protein per lane and 8%, 10%, or 12% SDS-PAGE gels. Proteins were subsequently wet-blotted onto nitrocellulose membranes (Biorad) and membranes were blocked for 1 hr using 5% non-fat dry milk in TBS-T. Immunoblot analysis was performed using the following primary antibodies: rabbit anti-ADAM10 (1:1000; polyclonal antiserum, B42.1), mouse monoclonal antibodies against PrP$^C$ (POM1 [for most of the experiments] or POM2 [for detection of shed PrP (*Figure 1E*)] (*Polymenidou et al., 2008*), 1:2500; A Aguzzi, Zurich, Switzerland), mouse monoclonal 1E4 (1:500; Sanquin, Amsterdam, The Netherlands), rabbit against p35/p25 (1:1000; mAB #2680; Cell Signaling), mouse Anti-Spectrin (1:500; MAB1622; Millipore, Bedford, MA), rabbit polyclonal antibody against calpain (H-240, 1:200; Santa Cruz Biotechnology, Santa Cruz, CA), and mouse monoclonal antibody against β-actin (1:2000; Millipore) or rabbit anti-Actin (1:2000; A5060; Sigma). Incubation with primary antibodies (diluted in 5% non-fat dry milk) was carried out overnight at 4°C on a shaking platform. Membranes were then washed 3× for 5 min with TBS-T, incubated for 45 min at room temperature with HRP-conjugated anti-mouse or anti-rabbit secondary antibodies (Promega, Madison, WI) diluted in 5% non-fat dry milk in TBS-T, and washed 6× for 5 min with TBS-T. The signal was detected after incubation of membranes with Pierce ECL or SuperSignal West Femto substrate (Thermo Scientific, Waltham, MA) using a CD camera imaging system (Biorad). If necessary, quantification of signal strengths was performed using QuantityOne software (Biorad) to measure the ratio of protein bands (glycotyping of PrP$^C$) or to determine relative protein expression against β-actin.

For assessment of PrP$^{Sc}$ levels and for use in inoculation experiments (Bioassay), 20% (wt/vol) homogenates of frontal brain were prepared in sterile phosphate buffer saline (PBS) without protease inhibitors. Again, samples were smashed 30× on ice using a Dounce homogenizer and subsequently spun down at 1000 rpm for 2 min. The resulting supernatant was either further diluted in PBS to yield a 1% homogenate used for inoculation of *tga20* reporter mice (see above) or 2–5 μl were digested with 20 μg/ml PK (Roche) in a total volume of 22 μl RIPA buffer for 1 hr at 37°C to assess the PrP$^{Sc}$ content. For detection of atypical prion fragments, selected samples were digested with 1 or 10 μg/ml or without PK for 1 hr at 37°C. 'Cold PK' treatment was performed with 200 μg/ml PK for 1 hr at 4°C. Digestion was stopped by adding 10× loading buffer and boiling for 6 min at 96°C. Subsequent SDS-PAGE and Western blot analysis was performed as described above with the exception that commercial Mini-PROTEAN TGX Any kD gels (Biorad) were used.

## Histological and immunohistochemical analysis

Morphological analysis was performed as described previously (*Altmeppen et al., 2011*; *Prox et al., 2013*). In brief, brains were dissected and fixed by immersion in 4% buffered formalin overnight. In the case of prion- or mock-inoculated animals, samples were inactivated for 1 hr in 98–100% formic acid before export from the facility. These samples were then incubated overnight with an excess of 4% buffered formalin. Samples were dehydrated and embedded in low melting point paraffin following standard laboratory procedures. Sections of 4 μm were prepared and either stained with hematoxylin and eosin (HE) following standard laboratory procedures or submitted to immunostaining following standard immunohistochemistry procedures using the Ventana Benchmark XT machine (Ventana,

Tucson, AZ). Briefly, deparaffinated sections were boiled for 30–60 min in 10 mM citrate buffer (pH 6.0) for antigen retrieval. All solutions were from Ventana. Sections were incubated with primary antibody in 5% goat serum (Dianova, Hamburg, Germany), 45% Tris buffered saline (TBS) pH 7.6, 0.1% Triton X-100 in antibody diluent solution (Zytomed, Berlin, Germany) for 1 hr. The following primary antibodies were used: monoclonal POM1 (*Polymenidou et al., 2008*) (1:100; Prof Dr Aguzzi, Zürich, Switzerland), SAF84 antibody against PrP$^C$ and PrP$^{Sc}$ (1:100; Cayman Chemicals, Ann Arbor, MI), anti-GFAP (1:200; M0761, DAKO, Hamburg, Germany), anti-Iba-1 (1:500; 019-19741, Wako Chemicals, Neuss, Germany), anti-NeuN (1:50; MAB377, Millipore). Detection was with anti-rabbit or anti-goat histofine Simple Stain MAX PO Universal immunoperoxidase polymer or Mouse Stain Kit (for detection of mouse antibodies on mouse sections). All secondary antibody polymers were purchased from Nichirei Biosciences (Tokyo, Japan). Detection of antibodies was performed with Ultra View Universal DAB Detection Kit or Ultra View Universal Alkaline Phosphatase Red Detection Kit from Ventana according to standard settings of the machine. Experimental groups were stained in one run, thereby providing identical conditions. The counterstaining was also performed by the machine according to common protocols. Additional negative controls included sections treated with secondary antibody only. For PrP$^{Sc}$ detection, mounted tissue sections (4 µm) were pretreated with 98% formic acid for 5 min. Further processing was performed on an automated staining machine (Benchmark XT, Ventana). Briefly, sections were pretreated with 1.1 mM sodium citrate buffer (2.1 mM Tris–HCl, 1.3 mM EDTA, pH 7.8) at 95°C for 30 min, digested with low concentration of PK for 16 min, incubated in Superblock for 10 min and then incubated with the PrP-specific antibody SAF 84 (see above), followed by secondary antibody treatment and detection.

## Electron microscopy

Small pieces of forebrain (2–3 mm³) of terminally prion-diseased A10 cKO mice and littermate controls were fixed in glutaraldehyde, postfixed in osmium tetroxide for 1–2 hr, dehydrated through a series of graded ethanols and propylene oxide, and embedded in Epon. Semi-thin sections were stained with toluidine blue. Ultrathin sections were stained with lead citrate and uranyl acetate, and specimens were examined using a JEM 100 C transmission electron microscope (JEOL, Tokyo, Japan). For the assessment of TVS, sample grids were divided into grid squares of equal size and the number of grid squares presenting TVS clusters was determined as published previously (*Falsig et al., 2012*).

## Isolation, cultivation, modification, and in vitro differentiation of NSCs

Adherently growing NSC cultures were established from the ganglionic eminence of 14-day-old wild-type and $^{Nestin}$A10 KO mice as described elsewhere (*Jorissen et al., 2010*; *Altmeppen et al., 2011*). In brief, we first established neurosphere cultures according to standard protocols (*Ader et al., 2001*; *Pressmar et al., 2001*). After two passages, neurospheres were enzymatically dissociated using Accutase (PAA Laboratories, Pasching, Austria) and cells were further cultivated in tissue culture flasks coated with 0.1% Matrigel (BD Biosciences, Franklin Lakes, NJ) in DMEM/F12 (Life Technologies, Carlsbad, CA) supplemented with 2 mM glutamine, 5 mM HEPES, 3 mM sodium bicarbonate, 0.3% glucose (all from Sigma–Aldrich, St Louis, MO; in the following termed NS medium) and 10 ng/ml epidermal growth factor (EGF), 10 ng/ml fibroblast growth factor-2 (FGF-2) (both from Tebu-Bio, Le-Perray-en-Yvelines, France), 1% N2 and 1% B27 (both from Life Technologies) to establish adherently growing NSC cultures (*Conti et al., 2005*; *Jung et al., 2013*). To express ADAM10 in $^{Nestin}$A10 cKO NSCs, the mouse *Adam10* cDNA was cloned into pcDNA3.1/Zeo(−) (Life Technologies), and the linearized plasmid was used to transfect $^{Nestin}$A10 KO NSC using the Nucleofector technology (Lonza, Basel, Switzerland) as described previously (*Richard et al., 2005*; *Altmeppen et al., 2011*). As control, $^{Nestin}$A10 KO NSCs were nucleofected with pcDNA3.1/Zeo(−) lacking the *Adam10* cDNA. To select for positive cells, cells were further cultivated in NS medium supplemented with 10 ng/ml EGF, 10 ng/ml FGF-2, 1% N2, 1% B27, and 200 µg/ml zeocin.

To induce neuronal differentiation of wild-type, $^{Nestin}$A10 KO and A10-nucleofected $^{Nestin}$A10 KO NSCs, cells were plated onto coverslips coated with 1% Matrigel and maintained for 4 days in NS medium supplemented with 5 ng/ml FGF-2, 1% N2, and 2% B27. Subsequently, cells were cultivated for another 4 days in a 1:1 mixture of NS medium and Neurobasal medium (Life Technologies) containing 0.25% N2 and 2% B27.

## Surface biotinylation assay in neuronally differentiated NSCs

After neuronal differentiation of NSCs in six-well plates, cells were washed 2× with cold PBS and incubated for 30 min with 0.5 mg EZ-Link Sulfo-NHS-SS-Biotin (Thermo Scientific) in PBS at 4°C under gentle agitation on a rocking platform. Cells were then washed 3× for 5 min at 4°C with 0.1% BSA in PBS and lysed with 500 µl RIPA buffer as described above. After centrifugation, supernatants were diluted 1:1 with Triton dilution buffer (100 mM TEA, 100 mM NaCl, 5 mM EDTA, 0.02% NaN$_3$, 2.5% Triton X-100, pH 8.6, +PI) and incubated for 1 hr with 200 µl pre-washed NeutrAvidin agarose beads (Thermo Scientific) at room temperature on a rotary wheel. Centrifugation was performed at 1000×$g$ for 1 min and supernatant was taken as 'lysate control'. Beads were washed 3× with wash buffer (20 mM TEA, 150 mM NaCl, 5 mM EDTA, 1% Triton X-100, 0.2% SDS, 0.02% NaN$_3$, pH 8.6, +PI) and spun down. Two more washing steps were performed with final wash buffer (20 mM TEA, 150 mM NaCl, 5 mM EDTA, pH 8.6, +PI) before adding 50 µl of 4× loading buffer including DTT and boiling for 6 min at 96°C to release biotinylated proteins from the beads. Supernatants were taken and loaded onto gels for SDS-PAGE as described above. 'Lysate controls' and biotinylated samples were detected via immunblotting using POM1 antibody. As reference and specificity control for surface biotinylated samples, membrane marker Flotillin-1 (murine purified anti-Flot-1; 1:1.000; BD Bioscience) was used.

## Immunofluorescence analysis

For surface staining of PrP$^C$ in neuronally differentiated NSCs, live cells were incubated with primary antibody against PrP$^C$ (POM1) for 1 hr at 4°C. Cells were then fixed in 4% paraformaldehyde in PBS (pH 7.4), blocked in PBS containing 0.1% bovine serum albumin and 0.3% Triton X-100 (both from Sigma), and incubated with polyclonal rabbit anti-β-tubulin III antibodies (Sigma) overnight at room temperature to identify nerve cells. Primary anti-PrP$^C$ and anti-β-tubulin III antibodies were detected with anti-mouse Cy2- and anti-rabbit Cy3-conjugated secondary antibodies (Jackson Immunoresearch Laboratories, West Grove, PA), respectively, and cell nuclei were stained with 4′,6-diamidino-2-phenylindole (DAPI; Sigma).

For MEF cells, surface staining of PrP$^C$ and ADAM10 was achieved by incubating live cells for 1 hr at 4°C with primary antibodies POM1 and monoclonal rat anti-mouse ADAM10 ectodomain antibody (1:100, R&D Systems, Minneapolis, MN), respectively. These antibodies were also applied after permeabilization of cells with 0.2% Triton X-100 in PBS. Goat anti-mouse IgG Alexa Fluor 488 and goat anti-rat IgG Alexa Fluor 568 (both from Life Technologies) were used as secondary antibodies prior to mounting the coverslips with DAPI Fluoromount-G (SouthernBiotech, Birmingham, AL) onto object slides.

## Quantitative RT-PCR

Cortical regions from A10 cKO mice (n = 5) and littermate controls (n = 5) were prepared and total RNA was isolated using the NucleoSpin RNAII kit (Macherey Nagel, Dueren, Germany) according to the manufacturer's instructions. 2 µg of DNAse-treated RNA was used for cDNA synthesis using the RevertAid cDNA Synthesis Kit (Thermo Fisher Scientific). Gene expression analysis for mouse PrP$^C$ and GAPDH were performed using the mouse Universal ProbeLibrary System (Roche Applied Science) for qRT-PCR experiments (PrP$^C$: 5′-GCCGACATCAGTCCACATAG, 5′-GGAGAGCCAAGCAGACTATCA, Probe: #71) on a LightCycler480 (Roche Diagnostics). PrP$^C$ gene expression levels were depicted as percentage of GAPDH expression using the ΔCT method for calculation. PCR efficiency of each assay was determined by serial dilutions of standards and these values were used for calculation.

## Statistical analysis

Statistical comparison of incubation times, Western blot quantifications, and qRT-PCR results between experimental groups was performed using Student's $t$-test for two-group comparisons with consideration of statistical significance at p values <0.05(*), <0.01(**), and <0.001 (***).

## Acknowledgements

We thank Dr Melanie Neumann and Kristin Hartmann from the Mouse Pathology Facility, University Medical Center Hamburg-Eppendorf (UKE), for their valuable support in the immunohistochemical stainings. Moreover, we thank Dr Bernd Zobiak and Dr Virgilio Failla from the UKE Microscopy and

Imaging Facility (UMIF) for technical support. We thank Prof Dr Adriano Aguzzi (Zurich, Switzerland) for providing us with the POM antibodies.

## Additional information

### Funding

| Funder | Grant reference number | Author |
| --- | --- | --- |
| Deutsche Forschungsgemeinschaft | SFB877 | Hermann C Altmeppen, Johannes Prox, Luise Linsenmeier, Paul Saftig, Markus Glatzel |
| Leibniz Association | Leibniz Graduate School | Hermann C Altmeppen |
| European Commission Directorate-General for Research and Innovation | REGPOT-2012-2013-1,7FP | Beata Sikorska, Pawel P Liberski |
| Werner Otto Stiftung | | Hermann C Altmeppen, Luise Linsenmeier |
| Hans & Ilse Breuer Stiftung | | Paul Saftig |
| Deutsche Forschungsgemeinschaft | GRK1459 | Paul Saftig, Markus Glatzel |
| Deutsche Forschungsgemeinschaft | FOR885 (GL 589/5) | Markus Glatzel |
| University of Hamburg | Young investigator research grant of the Medical Faculty | Hermann C Altmeppen |

The funders had no role in study design, data collection and interpretation, or the decision to submit the work for publication.

### Author contributions

HCA, Conception and design, Acquisition of data, Analysis and interpretation of data, Drafting or revising the article; JP, SK, BP, KK, FD, CB, AH, LL, BS, Acquisition of data, Analysis and interpretation of data, Drafting or revising the article; PPL, UB, Acquisition of data, Analysis and interpretation of data, Drafting or revising the article, Contributed unpublished essential data or reagents; PS, Conception and design, Analysis and interpretation of data, Drafting or revising the article, Contributed unpublished essential data or reagents; MG, Conception and design, Analysis and interpretation of data, Drafting or revising the article

### Author ORCIDs
Markus Glatzel, http://orcid.org/0000-0002-7720-8817

### Ethics

Animal experimentation: Our study was carried out in accordance with the principles of laboratory animal care (NIH publication No. 86-23, revised 1985) as well as the recommendations in the Guide for the Care and Use of Laboratory Animals of the German Animal Welfare Act on protection of animals. The protocol was approved by the Committee on the Ethics of the Freie und Hansestadt Hamburg—Amt für Gesundheit und Verbraucherschutz (permit number 48/09, 81/07 and 84/13). Every effort was made to minimize suffering.

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
