## [Decision Letter]

Thank you for sending your work entitled “The sheddase ADAM10 is a potent modulator of prion disease” for consideration at *eLife*. Your article has been favorably evaluated by Randy Schekman (Senior editor) and three reviewers, one of whom is a member of our Board of Reviewing Editors.

The Reviewing editor and the other reviewers discussed their comments before we reached this decision, and the Reviewing editor has assembled the following comments to help you prepare a revised submission.

Overall all three referees agree that this work is interesting and is in principle good material for *eLife*. However the work needs an in depth revision with regard to the conclusions made as they are not all supported by the provided data. Also rewriting the Results section and reorganisation of the way the data are presented is needed before the manuscript can be accepted. The referees are also concerned about the low number of animals used to support some of the conclusions.

Below follow the list of substantial comments that need to be fully addressed in a revised manuscript. We also urge the authors to address systematically the many other concerns of the referees as those will help to improve considerably this potentially very interesting manuscript.

Substantial comments:

1) The most important conclusion drawn by the authors is that suppression of PrP shedding by ADAM10 shortens the incubation time of the disease by ca. 30%, and therefore shedding of PrP by ADAM10 puts a brake on prion pathogenesis. This interpretation may be valid, but ADAM10 deficiency is highly pathogenic by itself, and the authors say that they couldn't even perform the experiment in CNS-specific conditional mutants because of embryonic lethality. It is therefore plausible (and indeed likely) that the accelerated death of prion-inoculated A10 cKO mice results nonspecifically from the combined pathology of prion infection plus absence of ADAM10. This possibility is rendered even more likely by the titer determinations which failed to demonstrate any difference in prion replication between A10 cKO and wt mice. The above is not necessarily a killer of this well-conducted study, but these alternative interpretations must by all means be discussed.

2) The claim that soluble PrP^C^ (generated by ADAM10) explains the spreading of the disease in the *tga20* (and the absence in ADAM) is only based on at best correlative evidence: the alleged reduced pathology in the ADAM may be due to reduced prion spread, but it may also pertain to secondary effects unrelated to prion spread (or to prion pathogenesis altogether), such as reduced processing of proinflammatory molecules, or impairment of neuron-microglia interaction by the lack of ADAM10. The authors have not directly investigated whether prion spread is modified in their mice (an admittedly difficult task, though not impossible). A knockout of ADAM10 in the *tga20* mice would allow investigating more precisely the role of cleaved PrP^C^ in the spreading of the disease in these mice. It might also be possible to inject /transduce /transgenically express soluble PrP^C^ in the ADAM10 cKO and rescue the spreading of the disease. As these experiments require substantial time, the authors might alternatively reconsider this conclusion and either tone down the interpretation of these findings, or at least mention the many possible alternative explanations.

3) In the Results section related to Figure 1, while mentioned in the Methods, the control for reintroducing ADAM10 cDNA into ADAM10 knockout cell lines (Lenti virus lacking ADAM10 cDNA) is not shown. It is important to show that the decrease in PrP^C^ signal is independent of the virus treatment itself. It would be desirable to have n>2 for quantification of surface PrP^C^ expression in A10 cKO+ADAM10.

4) The number of mice used for the crucial experiments in Figure 7 (incubation times) is meagre. I realize that there is little that can be done about this problem at this stage, but the authors would be well-advised to use numbers of mice appropriate for sufficient statistical power. The same applies to the titer determinations in *tga20* mice: just four recipient mice for each titration is pathetically low and will by necessity undermine the precision of the determination.

Other comments:

Reviewer #1:

1) In the Abstract a conclusion is made with regard to treatment options for other neurodegenerative diseases (apart from prion). In the absence of any data, I suggest to remove this claim from the Abstract and only mention this in the Discussion as an interesting speculation.

2) Figures 1 and 2: the paper contains a whole series of experiments (covering two full figures) to proof that in the absence of ADAM 10, prion expression is increased and shedding is decreased. I believe that these data should be summarized in one figure, especially because this is not entirely novel. Moreover, instead of many different data it is better to have a couple of experiments fully documented. For instance it is not clear what was quantified in Figure Panel 1A and this should be complemented with the blot or the figure on which the quantitation is based. Panel 1C should include for the ADAM10 KO a similar analysis as for the control with regard to PrP surface staining. In Figure 2: the immunohistochemistry shows a particular strong staining in the mouse brains of the ADAM cKO, more than what is seen in the western blot and expression is very diffuse over all cells (while the ko is only in a subset of neurons). Additional controls e.g. immunostaining of the cerebellum and more precise details of the cellular stainings would help to understand this better and to demonstrate for instance that the slides of wild type and knock out were incubated for similar length of time. It is also important to show that PrP accumulation is only seen in the neurons where ADAM10 was knocked out, as this is an important point also with regard to the conclusions with regard to the propagation of the disease.

3) The very strong conclusion in the second paragraph of the Discussion: “and show for the first time that PrP^C^ and ADAM10 co-localize at the plasma membrane” is based on a single piece of weak evidence (inset in Panel 1C). The resolution of this type of experiment is insufficient to make such a strong claim.

4) In the ninth paragraph of the Results section: “Taken together, these data indicate that at terminal prion disease, lack of PrP^C^-shedding does not influence local PrP^Sc^ distribution while it does impact the appearance of TVS”. This observation is mentioned here, and not further discussed or explained. Why mention it?

5) Figure 5: I find it puzzling that there is such a contrast in the accumulation of proteinase K resistant species in the ADAM10 ko versus the *tga20* mouse? In principle both animals have more uncleaved prion protein on the cell membrane. Thus the main difference is the shed prion in the *tga20* mouse. Would one conclude from this that the shed prion inhibits the generation of proteinase K resistant forms of the PrP^Sc^? As also indicated above, I think that it is necessary to study the ADAM10 KO effect in the *tga20* mouse to allow making real conclusions with regard to the precise effect of ADAM10.

6) Figure 5: The activation of calpain is linked to membrane pores in the text. I suppose that it is linked to Calcium influx? This needs to be better formulated.

Reviewer #2:

1) In its current form the manuscript has a few areas of weakness that need to be improved. In particular, throughout the manuscript the PrP^C^ and PrP^Sc^ levels, the incubation periods, and the spongiosis levels are compared between different mouse lines (A10 cKO, wild-type, *tga20*), different time points (65d, 95d), and different brain regions (forebrain, cerebellum, whole brain), but at present these comparisons are a bit haphazard. It would be great to include a table to assemble all these scattered data into one location. This table would make it easier for the reader to keep track of these differences and better follow the arguments the manuscript makes.

2) Throughout the manuscript a comma is used as the decimal mark. For proper English a full stop should be used instead.

3) In several places the A10 cKO results are compared with those that were obtained with mice expressing anchorless PrP^C^ (e.g. at the end of the third paragraph of the Introduction, eighth paragraph of the Discussion, etc.). At present the comparison is limited to an anchorless PrP^C^ low-expressor mouse line (28), but it would be desirable to also include high-expressor results into this discussion (Stohr et al., 2011). Particularly, since the latter animals express PrP at levels closer to the mouse lines that are used in this study.

4) The first paragraph of the Results section describes results from a complete ADAM10 knockout mouse line, according to the figure legend the corresponding figures (Figure 1) show only A10 cKO data. This mismatch needs to be cleared up.

5) It is difficult to judge the alleged co-localization of PrP^C^ and ADAM10 (Figure 1), since the magnification is so low. Higher resolution images would be needed to make this a convincing argument.

6) The electron micrographs in Figure 4 show tubulovesicular structures (TVS), which are claimed to be increased in diseased A10 cKO mice. It would be worthwhile to document and quantify this increase, since TVS and their link to prion disease are controversial. Ultimately, the A10 cKO mice may help to demystify these structures and their role in the prion diseases.

7) The comparison of PrP^Sc^ levels between different mouse lines is solely based on PK-resistance (Figure 5). While this surrogate marker is widely used to estimate prion infectivity levels, the possible presence of protease-sensitive forms of PrP^Sc^ should be taken into consideration and discussed.

8) In the conclusions, ADAM10 is being promoted as a potential therapeutic target for prion diseases. The finding that a reduction of ADAM10 leads to a shortening of the incubation period, while its normal function increases the spreading of PrP^Sc^, suggests a no-win scenario. Hence, a more refined and discerning discussion would be needed to support such a statement.

9) Lastly, grammatical errors can be found all the way through the manuscript, a thorough proofreading would benefit the readability of the text.

Reviewer #3:

1) In Figure 4, WB analysis of protein expression was compared only in frontal brain homogenates. This needs to be controlled by including whole brain homogenates as well.

2) In panel C, the GFAP signal of CD1 mock ADAM10 cKO seems higher than the control. Is this image representative? If yes, is there an explanation for this?

3) In panel D: How was the quantification done and what is the relevance of higher numbers of tubulovesicular structures in ADAM10 cKO? Does this bear relevance for the paper at all?

4) Results related to Figure 5: It would be important to show that the increase of PrP^Sc^ levels in forebrain homogenates of ADAM10 cKO is comparable to the overall PrP^Sc^ content of wild type mouse brains. This could underline that the difference resides in the lack of shedding and spreading of PrP^Sc^, leading to local accumulation of PrP^Sc^ in ADAM10 cKO. For this, whole brain homogenates should be compared.

5) As for the results shown in Figure 6, the comparison of infectivity tests of non-forebrain regions of control and ADAM10 cKO would be important, as it could identify any infectious but non-toxic PrP subpopulations which still might leave PrP^Sc^ aggregates in ADAM10 cKO.

6) Discussion: The authors state that PrP “is a receptor mediating neurotoxicity in common neurodegenerative proteinopathies such as Alzheimer's disease”. This is a highly controversial contention propagated by Dr. Strittmatter, and there is more than half-dozen papers out there showing that this idea is, at best, limited to certain animal models. It is the authors' privilege to choose what they wish to believe, and I understand that the charitable interpretation of the Strittmatter paper might add to the perceived importance of the current study, but in fairness they should at least mention the evidence against PrP^C^ being the Aβ receptor.

7) One problematic point is the following: It is stated that ADAM10 cKO mice show massive neurodegeneration in the forebrain, but not in the cerebellum and brainstem, while controls including *tga20* show diffuse neurodegeneration. If this is really true, how can one explain completely identical clinical phenotypes?

8) As for Figure 5, the total amount of calpain is most certainly irrelevant to neurodegeneration, since calpains are mostly inactive in the absence of raised intracellular calcium. What matters is the amount of activated calpains. This can be assessed by investigating the amount of cleavage of a calpain substrate, such as spectrin (alpha-fodrin). The mere detection of calpains by Western blot is irrelevant. The whole argument about calpains needs to be revisited by activity measurements, or else it should be deleted in toto. This becomes painfully evident by the authors' finding that on the one hand there is prion toxicity (allegedly through calpain) with profound neurodegeneration in the forebrain of ADAM10 cKO, but on the other hand the significantly faster disease in *tga20* mice goes along with significantly less calpain Also, a comparison of overall (not only forebrain) calpain activity levels to mock inoculated animals could help.

9) When mentioning the possible inhibition of PrP^Sc^ formation by shedded PrP^C^, it would be desirable to prove this statement with PMCA + shed PrP^C^.

10) In Figure 1, POM2 is named as the applied antibody, while the Methods section describes POM1 for this purpose.

---

## [Author Response]

*1) The most important conclusion drawn by the authors is that suppression of PrP shedding by ADAM10 shortens the incubation time of the disease by ca. 30%, and therefore shedding of PrP by ADAM10 puts a brake on prion pathogenesis. This interpretation may be valid, but ADAM10 deficiency is highly pathogenic by itself, and the authors say that they couldn't even perform the experiment in CNS-specific conditional mutants because of embryonic lethality. It is therefore plausible (and indeed likely) that the accelerated death of prion-inoculated A10 cKO mice results nonspecifically from the combined pathology of prion infection plus absence of ADAM10. This possibility is rendered even more likely by the titer determinations which failed to demonstrate any difference in prion replication between A10 cKO and wt mice. The above is not necessarily a killer of this well-conducted study, but these alternative interpretations must by all means be discussed*.

The mice we used for the in vivo experiments are CNS-specific conditional mutants with Cre-mediated deletion of ADAM10 occurring in forebrain neurons under the control of the *Camk2a* promoter. These mice display an almost normal life span once they have survived a critical weaning period, additionally they do not show any gross morphological abnormalities in the CNS (Prox et al., 2013a).

The ADAM10 KO mice showing perinatal lethality, the reviewing editor is referring to, were only used for the ex vivo experiments shown in Figure 1 (determination of PrP surface levels in neuronally differentiated neural stem cells (Figure 1) and analyses of PrP-shedding in primary neurons derived from embryonal day 14 mice (Figure 1). In these mice, ADAM10 is deleted under the control of the Nestin promoter with ADAM10 deficiency occurring in progenitor cell types present in the neuroectoderm, the developing mesonephros, and the somites (46). We have clarified this in the first paragraph of the Results section in the manuscript:

“To further investigate this we generated neural stem cells (NSCs) from embryonic day (E) 14 wild-type and ADAM10 conditional knockout mice with an inactivation of the *Adam10* gene in neuroectodermal progenitor cells (^Nestin^A10 KO mice) (46).”

Nevertheless, we agree with the reviewing editor that effects on prion disease acceleration caused by lack of ADAM10 in forebrain neurons (^*Camk2a*^ADAM10 cKO) may not solely be caused by lack of ADAM10-mediated PrP shedding and increased neuronal PrP^C^ levels. We have thus introduced a text fragment in the fourth paragraph of the Discussion section where we discuss this possibility in detail, with special emphasis on potential contribution by ADAM10 substrates of the immune system:

“We considered the possibility that non-PrP dependent effects resulting from ADAM10 deficiency may have contributed to disease acceleration. […] Finally, we consider that increased PrP^Sc^ levels found in A10 cKO mice strongly argue in favor of PrP-dependent effects of ADAM10 depletion.”

Regarding the observed identical prion titer, we would like to clarify that titers of prion infectivity and presence of potentially neurotoxic PrP conformers are rather not congruent. In fact, a growing body of published reports indicates a clear dissociation between these. We have clarified and referenced this in paragraph seven of the Discussion section. As suggested by Reviewer #2 (point 7), we have also checked for the presence of aberrant prion species/fragments that could contribute to the uncoupling of infectivity and PrP^Sc^ amounts yet did not find any evidence for those in our samples (at 95 dpi) as shown in new Figure 6—figure supplement 1.

“A10 cKO mice and controls showed identical prion titers […] while it does not influence production of ‘prions’.”

*2) The claim that soluble PrP*^*C*^
*(generated by ADAM10) explains the spreading of the disease in the* tga20 *(and the absence in ADAM) is only based on at best correlative evidence: the alleged reduced pathology in the ADAM may be due to reduced prion spread, but it may also pertain to secondary effects unrelated to prion spread (or to prion pathogenesis altogether), such as reduced processing of proinflammatory molecules, or impairment of neuron-microglia interaction by the lack of ADAM10. The authors have not directly investigated whether prion spread is modified in their mice (an admittedly difficult task, though not impossible). A knockout of ADAM10 in the* tga20 *mice would allow investigating more precisely the role of cleaved PrP*^*C*^
*in the spreading of the disease in these mice. It might also be possible to inject /transduce /transgenically express soluble PrP*^*C*^
*in the ADAM10 cKO and rescue the spreading of the disease. As these experiments require substantial time, the authors might alternatively reconsider this conclusion and either tone down the interpretation of these findings or at least mention the many possible alternative explanations.*

We agree with the reviewing editor that experiments directly investigating if lack of ADAM10 mediated shedding of PrP is responsible for the reduction of prion spread in our mice are difficult if not impossible to conduct. Although the proposed experiment of crossing ADAM10 cKO into *tg*a*20* mice or transgenic expression of soluble PrP^C^ on an ADAM10 cKO background will likely provide important information on the correlation between shedding of PrP and spread of prion pathology, this will rather not get rid of non-PrP mediated effects. Additionally, we believe that presence of soluble PrP^C^ will introduce another variable by potential interference with the PrP^C^-PrP^Sc^ conversion process, as discussed in paragraph eight of the Discussion. Nevertheless, we are initiating these breedings which are, as stated by the reviewing editor, time consuming.

We also agree that our evidence at this point is merely correlative. Therefore, we have toned down our interpretation in the Results and Discussion sections and specifically mention secondary effects unrelated to prion spread:

“ADAM10-mediated shedding might contribute to spreading of prion pathology” (in the Results section) and “Non-PrP dependent effects resulting from ADAM10 […] efficiency of prion spread is possible (35; 38)” (Discussion section).

*3) In the Results section related to*
Figure 1*, while mentioned in the Methods, the control for reintroducing ADAM10 cDNA into ADAM10 knockout cell lines (Lenti virus lacking ADAM10 cDNA) is not shown. It is important to show that the decrease in PrP*^*C*^
*signal is independent of the virus treatment itself. It would be desirable to have n>2 for quantification of surface PrP*^*C*^
*expression in A10 cKO+ADAM10*.

We are thankful for making us aware of this oversight. We have now included the data from the empty vector control. As stated in the Method section, we used Nucleofector® technology instead of lenitiviral transduction (this term was mistakenly included in the previous version) to genetically introduce ADAM10 or the linearized vector only. Moreover, we have performed additional surface biotinylation experiments to increase “n” numbers of independent culture samples and now show representative Western blots (Figure 1). We also performed a new set of immunofluorescent stainings including the requested empty vector control (Figure 1). The new data are mentioned in the Results section (first paragraph) as well as in the figure legend for Figure 1.

*4) The number of mice used for the crucial experiments in*
Figure 7
*(incubation times) is meagre. I realize that there is little that can be done about this problem at this stage, but the authors would be well-advised to use numbers of mice appropriate for sufficient statistical power. The same applies to the titer determinations in* tga20 *mice: just four recipient mice for each titration is pathetically low and will by necessity undermine the precision of the determination*.

We are aware of our shortcomings in this respect and while it was not possible to increase the amount of A10 cKO and littermate controls for time reasons and limitations in breeding capacities, we have been able to perform new experiments to increase the number of *tg*a*20* reporter mice for prion titer determination. For the most interesting and important time point (95 dpi) we have inoculated more indicator mice per group (from four to six) and we have bioassayed additional samples (one additional inoculum for both A10 cKO and littermate control, into n=6 reporter mice). The data remain largely unchanged but have now become more robust. This is mentioned in the figure legend for Figure 7 as well as in the Results and Methods sections.

*Other comments*:

Reviewer #1:

*1) In the Abstract, a conclusion is made with regard to treatment options for other neurodegenerative diseases (apart from prion). In the absence of any data, I suggest to remove this claim from the Abstract and only mention this in the Discussion as an interesting speculation*.

We understand the point of this reviewer and followed the advice and have removed this claim from the Abstract and carefully speculate on this interesting issue in the Discussion.

*2)*
Figures 1 and 2*: the paper contains a whole series of experiments (covering two full figures) to proof that in the absence of ADAM 10, prion expression is increased and shedding is decreased. I believe that these data should be summarized in one figure, especially because this is not entirely novel. Moreover, instead of many different data it is better to have a couple of experiments fully documented. For instance it is not clear what was quantified in Figure Panel 1A and this should be complemented with the blot or the figure on which the quantitation is based. Panel 1C should include for the ADAM10 KO a similar analysis as for the control with regard to PrP surface staining. In*
Figure 2*: the immunohistochemistry shows a particular strong staining in the mouse brains of the ADAM cKO, more than what is seen in the western blot and expression is very diffuse over all cells (while the ko is only in a subset of neurons). Additional controls e.g. immunostaining of the cerebellum and more precise details of the cellular stainings would help to understand this better and to demonstrate for instance that the slides of wild type and knock out were incubated for similar length of time. It is also important to show that PrP accumulation is only seen in the neurons where ADAM10 was knocked out, as this is an important point also with regard to the conclusions with regard to the propagation of the disease*.

We thank this reviewer for the helpful comments on Figures 1 and 2. In light of the fact that Reviewers #2 and #3 as well as the reviewing editor commented on Figure 1, and given that almost all of the data are novel and, as we consider, relevant to the rest of the manuscript, we decided to keep revised versions of both figures. We are convinced that Figure 1 is a reasonable start into the overall theme of this paper by revealing effects of ADAM10 depletion on PrP^C^ levels in different cellular lineages and thus generalizing those mechanisms. Although we previously reported lack of PrP^C^ shedding in ^Nestin^ADAM10 KO and increased shedding by *tg*a*20* neurons, we also tend to keep the hitherto unpublished Figure 1 to support our arguments with regard to inefficient PrP^Sc^ production and potentially increased prion spread in infected *tg*a*20* mice as described in the Discussion. With Figure 2 we then introduce the novel *Camk2a* ADAM10 cKO mouse model with special emphasize on its PrP^C^ expression pattern.

Following the suggestions of this reviewer we now:

a) Show representative blots (including the empty vector control) and the ADAM10 expression for PrP^C^ membrane levels experiment in neuronally differentiated neural stem cells (Figure 1);

b) Provide new images for the PrP^C^ surface staining including the empty vector control cells (Figure 1);

c) Show detailed, higher resolution images of the colocalization between ADAM10 and PrP^C^ on the plasma membrane of control MEF and also included this magnification for the ADAM10 KO MEF (Figure 1);

d) Include a more detailed panel of the PrP immunohistochemistry of control and ADAM10 cKO brains with higher magnifications now also showing details of the cerebellum (Figure 2). Moreover, we provide a new figure with co-staining of a neuronal marker (NeuN) and PrP (Figure 2—figure supplement 1) to demonstrate the regional variation in PrP immunohistochemistry in the brain and differences between ADAM10 cKO mice and wild-type littermates. This analysis reveals that the signal in PrP immunohistochemistry is increased in regions where ADAM10 is deleted thus conforming previous data on the *Camk2a* driven deletion strategy ([24]; Prox et al., 2013a). This is mentioned in the Results (fifth paragraph) as follows:

“In addition, a coronal brain section showing co-staining of PrP^C^ with the neuronal marker NeuN correlates with the *Camk2a* driven ADAM10 knockout strategy (24) by showing increased PrP^C^ expression in hippocampal and cortical areas as well as in the striatum (Figure 2—figure supplement 1).” Moreover, this is now also illustrated for the prospective readers' convenience and orientation in a new scheme (Figure 3).

The new figures are described in the figure legends. Additionally, we have added experimental details on the highly standardized immunostaining with an automated system guaranteeing identical incubation times and staining conditions (“Experimental groups were stained in one run thus providing identical conditions”, paragraph nine of Materials and methods section).

With regard to the last point we agree that our immunohistochemical staining for PrP^C^ shows a rather general staining yet with differences in intensities between genotypes. We aimed at showing that increased PrP^C^ staining is restricted to neurons depleted of ADAM10. Since neurons are highly branched cells and PrP is highly expressed on the neuronal cell surface, single neuron analysis in 4 μ m thick paraffin sections was not feasible. Moreover, all other cell types in the CNS (though not affected by the knockout strategy) contribute to the overall staining. Finally, to the best of our knowledge there is no published reliable antibody against ADAM10 working in immnunohistochemical staining of mouse brain sections. We have nevertheless tried different ADAM10 antibodies (rabbit anti-ADAM10 (B42.1); rabbit anti-ADAM10 amino acids 608-627 (kindly provided by Dr. A. Chalaris, Kiel); rat anti-ADAM10 ectodomain antibody (MAB946)) under different staining conditions in formalin-fixed and cryo-fixed brain samples yet were not able to detect specific signals. Thus, we could not perform co-stainings of ADAM10 and PrP^C^ which would help to answer this point of the reviewer.

*3) The very strong conclusion in the second paragraph of the Discussion: “and show for the first time that PrP*^*C*^
*and ADAM10 co-localize at the plasma membrane” is based on a single piece of weak evidence (inset in Panel 1C). The resolution of this type of experiment is insufficient to make such a strong claim*.

We agree with this reviewer, have provided higher resolution figures, have deleted this strong claim and modified the sentence. Moreover, we became aware of a report published in the meantime also showing colocalization of ADAM10 and PrP^C^. We have now added this reference (Chen et al.,, 2014) to the second paragraph of the Discussion section.

*4) In the ninth paragraph of the Results section: “Taken together, these data indicate that at terminal prion disease, lack of PrP*^*C*^*-shedding does not influence local PrP*^*Sc*^
*distribution while it does impact the appearance of TVS”. This observation is mentioned here, and not further discussed or explained*. *Why mention it?*

We have to admit that TVS are not in the focus of our study and thus we did not discuss these structures in great detail. However, we think that the observation of increased TVS amounts in our model might be interesting to the prion field. Since Reviewer #2 suggested a more detailed workup of TVS including a quantification of these structures, we decided to keep these data including a quantification of TVS as a supplementary figure (Figure 5—figure supplement 2). Additionally, we have slightly extended the Results section providing a description and references on TVS in prion disease (paragraph eleven) and added experimental details (eleventh paragraph of Materials and methods):

“Remarkably, prion-diseased ADAM10 cKO mice presented with extraordinarily high abundance of tubulovesicular structures (TVS) (Figure 5—figure supplement 2) (45; 61). These are spherical structures of approx. 25-37 nm in diameter that are specific for prion diseases though devoid of PrP (45; 61; 62). Thus, our model may allow purification of these structures and could contribute to unravel the nature and relevance of TVS in prion diseases.”

*5)*
Figure 5*: I find it puzzling that there is such a contrast in the accumulation of proteinase K resistant species in the ADAM10 ko versus the* tga20 *mouse? In principle both animals have more uncleaved prion protein on the cell membrane. Thus the main difference is the shed prion in the* tga20 *mouse. Would one conclude from this that the shed prion inhibits the generation of proteinase K resistant forms of the PrP*^*Sc*^*? As also indicated above, I think that it is necessary to study the ADAM10 KO effect in the* tga20 *mouse to allow making real conclusions with regard to the precise effect of ADAM10*.

We thank Reviewer #1 for the advice to better clarify this point. In fact, the suggested conclusion of this reviewer is exactly what we think. In addition to our analysis of PrP^C^ surface levels of neuronally differentiated neural stem cells derived from ADAM10 cKO and littermate control mice (showing an approx. 1.6-fold increase when ADAM10 is lacking; Figure 1), we now also analyzed PrP^C^ membrane levels in primary neurons of *tg*a*20* and C57BL/6 control mice (Figure 10).Author response image 1.Increased PrP^C^ surface levels in primary neurons of *tg*a*20* mice. Western blot analysis of lysates (on the left) and biotinylated surface proteins (on the right) of primary neurons from E14 embryos of *tg*a*20* and C57BL/6 wild-type mice. Flotillin served as a loading control. Quantification of densitometric analysis of surface PrP^C^ is shown below. C57BL/6 set to 1 (+/-0.1 SEM); *tg*a*20*: 3.1 +/-0.5; n=6 samples per genotype from 2 independent experiments; **p=0,007).

Our analysis revealed that *tg*a*20* mice (which overexpress PrP^C^ approx. 7-fold (Fischer, 1996)) show a ∼3-fold increase in PrP^C^ surface levels. Thus, while we think that PrP^C^ membrane levels are decisive for prion-associated neurotoxicity and disease duration (with *tg*a*20* showing much shorter incubation times than ADAM10 cKO mice), elevated total as well as surface levels of PrP^C^ (*tg*a*20* > ADAM10 cKO > wild-type controls) do not explain the differences in PrP^Sc^ formation. In fact, we think that shed PrP^C^ inhibits the conversion to PrP^Sc^ in vivo. As discussed in the main text (Discussion section) there is also some evidence from other studies showing that anchorless versions of PrP^C^ act as antagonists of the conversion process. We also added a reference showing this inhibitory effect in PMCA approaches (49). Soluble PrP^C^ generated by ADAM10-mediated shedding could represent the physiological correlate of this. We speculate that lack of the protease is responsible for efficient PrP^Sc^ production in ADAM10 cKO mice whereas increased production of shed PrP^C^ in *tg*a*20* mice (partially) explains their poor conversion rate (a known phenomenon of the latter mouse model that, to our knowledge, has not been solved to date). Fitting to this concept, PrP^Sc^ production was decreased in mice moderately overexpressing ADAM10 (31). We have newly added this reference in this context in the main text (paragraph six of the Discussion section). As mentioned in our response to the reviewing editor (major point 2) we agree that the crossing of ADAM10 cKO and *tg*a*20* mice is an interesting experiment to gain deeper insight.

*6)*
Figure 5*: The activation of calpain is linked to membrane pores in the text. I suppose that it is linked to Calcium influx? This needs to be better formulated*.

We have realized that our workup and statements regarding the membrane pore model was not clearly formulated and experimentally addressed. This point is addressed in detail below (Reviewer #3, point 8). To answer this specific point: Yes, if correct, our model would suggest that membrane pore formation leads to Ca^2+^ influx. This is now clarified in the Discussion (eighth paragraph) and in the figure legend of Figure 9.

Reviewer #2:

*1) In its current form the manuscript has a few areas of weakness that need to be improved. In particular, throughout the manuscript the PrP*^*C*^
*and PrP*^*Sc*^
*levels, the incubation periods, and the spongiosis levels are compared between different mouse lines (A10 cKO, wild-type,* tga20*), different time points (65d, 95d), and different brain regions (forebrain, cerebellum, whole brain), but at present these comparisons are a bit haphazard. It would be great to include a table to assemble all these scattered data into one location. This table would make it easier for the reader to keep track of these differences and better follow the arguments the manuscript makes*.

We thank this reviewer for the advice of providing a better structure of presented data. For this purpose we have prepared a new Figure (Figure 3) aiming to clarify the distribution of ADAM10 depletion within the brain and to explain our experimental strategy (Figure 3). As requested by this reviewer we have also included a table (Figure 3) summarizing our most important findings with comparison of prion relevant changes for different genotypes, brain regions and time points. Moreover, this figure provides direct reference to the relevant figures presenting original data. We hope that Figure 3 improves understanding of our study and makes it more convenient for readers to follow our arguments. The figure is mentioned in the Results section (paragraphs six and seven) as follows:

“Based on previous reports ([24]; Prox et al., 2013a) […] chosen for immunohistochemical or biochemical analyses.”

and:

“For better orientation, Figure 3 […] showing the original data described below.”

*2) Throughout the manuscript a comma is used as the decimal mark. For proper English a full stop should be used instead*.

We have now corrected this and use a full stop instead.

*3) In several places the A10 cKO results are compared with those that were obtained with mice expressing anchorless PrP*^*C*^
*(e.g. at the end of the third paragraph of the Introduction, eighth paragraph of the Discussion, etc.). At present the comparison is limited to an anchorless PrP*^*C*^
*low-expressor mouse line (*[28]*), but it would be desirable to also include high-expressor results into this discussion (Stohr et al., 2011). Particularly, since the latter animals express PrP at levels closer to the mouse lines that are used in this study*.

We have mentioned these mice and referenced this study in the Introduction (end of the third paragraph).

“High expression of anchorless PrP^C^ leads to formation of prions and a late onset neurological disease (Stohr et al., 2011).”

We also think that it is a good suggestion to mention these mice since this study contributes to our understanding of roles played by anchorless PrP. However, we have refrained from an in depth discussion of these mice since spread of prion pathology in the CNS was not the focus of that study.

*4) The first paragraph of the Results section describes results from a complete ADAM10 knockout mouse line, according to the figure legend the corresponding figures (*Figure 1*) show only A10 cKO data. This mismatch needs to be cleared up*.

We are thankful for this hint. Indeed, Figure 1 show data derived from murine embryo fibroblasts (MEF) of a complete ADAM10 knockout (KO) mouse line. We have corrected this in the figure legend and in the labelling of the figure itself. Moreover, the term “ADAM10 cKO” (or “A10 cKO”) is now only used when referring to *Camk2a*-Cre ADAM10 conditional knockout mice.

*5) It is difficult to judge the alleged co-localization of PrP*^*C*^
*and ADAM10 (*Figure 1*), since the magnification is so low. Higher resolution images would be needed to make this a convincing argument*.

As mentioned in our response to Reviewer #1, we now provide higher resolution images of both wild-type control and ADAM10 KO murine embryonic fibroblasts.

*6) The electron micrographs in*
Figure 4
*show tubulovesicular structures (TVS), which are claimed to be increased in diseased A10 cKO mice. It would be worthwhile to document and quantify this increase, since TVS and their link to prion disease are controversial. Ultimately, the A10 cKO mice may help to demystify these structures and their role in the prion diseases*.

As mentioned in our response to Reviewer #1, we did not put a special focus on TVS in our study. Nevertheless, our observation of high abundance of these structures in terminally prion diseased ADAM10 cKO mice (compared to wild-type controls in this study as well as to prion disease models published elsewhere) might be of interest to the prion field. Thus, we decided to devote an entire supplementary figure to the topic of TVS (Figure 5—figure supplement 2). We now also provide quantification, a short description on TVS and more references (paragraphs eleven and twelve of the Results section). We agree with this reviewer that our model “may help to demystify these structures” and ultimately solve their true significance. However, we are fully aware that this would require a much more refined way of analysis.

With regard to Figure 5 (former Figure 4) we would like to clarify that we introduced a new EM picture showing vacuolization with membranous structures (A). In addition, we modified the scale bars (in the figure and figure legends of B and C) as we became aware of a mistake in the former version of this manuscript.

*7) The comparison of PrP*^*Sc*^
*levels between different mouse lines is solely based on PK-resistance (*Figure 5*). While this surrogate marker is widely used to estimate prion infectivity levels, the possible presence of protease-sensitive forms of PrP*^*Sc*^
*should be taken into consideration and discussed*.

Since our data, as discussed above, support an uncoupling of PrP^Sc^ amounts and prion infectivity, we followed the suggestion of this reviewer and analyzed our samples for the presence of aberrant, low-abundance or PK sensitive prion forms/fragments. However, we did not find any atypical banding following digestions with “cold PK” (i.e. digestion with 200 µg/ml PK at 4°C for 1 h) or PK dilutions and detection with PrP-specific 1E4 or POM1 antibodies. This is now shown in the new Figure 6—figure supplement 1 and mentioned in the Methods (under the subsection “Western blot analysis”) and in the subsection “Enhanced PrP^Sc^ formation and calpain levels in A10 cKO mice” in Results:

“There were no atypical PrP patterns as assessed by PK digestion at 4°C (‘cold PK’) or with lower dilutions of PK (Figure 6—figure supplement 1).”

Moreover, we mentioned this issue in the revised Discussion (seventh paragraph):

“A10 cKO mice and controls showed identical prion titers, yet elevated levels of PrP^Sc^ and no evidence for atypical protease-sensitive forms of PrP^Sc^.”

*8) In the conclusions, ADAM10 is being promoted as a potential therapeutic target for prion diseases. The finding that a reduction of ADAM10 leads to a shortening of the incubation period, while its normal function increases the spreading of PrP*^*Sc*^*, suggests a no-win scenario. Hence, a more refined and discerning discussion would be needed to support such a statement*.

We agree with this reviewer that the dual role of ADAM10 in prion disease makes therapeutic approaches questionable. Thus, rather than embarking on a lengthy discussion of this topic we preferred to delete this claim in the conclusion of the Discussion (tenth paragraph):

“Understanding this dual role of ADAM10 in prion disease brings together current concepts of prion biology and might reveal mechanistic insight into important pathophysiological processes.”

*9) Lastly, grammatical errors can be found all the way through the manuscript, a thorough proofreading would benefit the readability of the text*.

All co-authors have checked the final manuscript again carefully and we have engaged a professional proofreader to eliminate errors.

Reviewer #3:

*1) In*
Figure 4*, WB analysis of protein expression was compared only in frontal brain homogenates. This needs to be controlled by including whole brain homogenates as well*.

As demonstrated in the new Figure 3, we had chosen to take samples from frontal brain for biochemical analysis while the rest of the brain was taken for in-depth immunohistochemical (IHC) analysis. Thus, unfortunately we are unable to present the requested controls. However, our IHC analysis of PK-resistant material (now shown in Figure 5) shows similar staining pattern indicating that equal amounts of PrP^Sc^ exist at a terminal state of disease in ADAM10 cKO (at ∼103 dpi) and littermate controls (at ∼146 dpi).

2) In panel C, the GFAP signal of CD1 mock ADAM10 cKO seems higher than the control. Is this image representative? If yes, is there an explanation for this?

This reviewer has correctly noticed that ADAM10 cKO mice indeed do show a mild gliosis. This has been published before (Prox et al., 2013a) and is indeed representative yet there is no one obvious explanation for this given the multitude of ADAM10 substrate in the brain discussed above. However, in both genotypes a much more dramatic increase in GFAP signal can be observed upon challenge with prions (Figure 5).

3) In panel D: How was the quantification done and what is the relevance of higher numbers of tubulovesicular structures in ADAM10 cKO? Does this bear relevance for the paper at all?

As requested by this reviewer and already mentioned in our response to reviewers #2 and #3 we now provide quantification, description and references on TVS and more experimental detail (twelfth paragraph of the Results section). Although we are, at this stage, unable to explain the role of increased TVS abundance in ADAM10 cKO mice (which also show increased PrP^Sc^ production), we think that this observation might have some relevance to the prion field. As suggested by Reviewer #2 a detailed analysis of these structures “may help to demystify these structures and their role in the prion diseases” in the future. However, since TVS are not the main focus of the current study, we now present this data in a supplementary figure (Figure 5—figure supplement 2).

*4) Results related to*
Figure 5*: It would be important to show that the increase of PrP*^*Sc*^
*levels in forebrain homogenates of ADAM10 cKO is comparable to the overall PrP*^*Sc*^
*content of wild type mouse brains. This could underline that the difference resides in the lack of shedding and spreading of PrP*^*Sc*^*, leading to local accumulation of PrP*^*Sc*^
*in ADAM10 cKO. For this, whole brain homogenates should be compared*.

We have to admit that this would be an interesting comparison. However, as stated in our response to point 1 and shown in Figure 3 we did not consider taking these samples for biochemical analysis when we planned our experimental strategy but rather decided for a detailed immunohistochemical assessment.

*5) As for the results shown in*
Figure 6*, the comparison of infectivity tests of non-forebrain regions of control and ADAM10 cKO would be important, as it could identify any infectious but non-toxic PrP subpopulations which still might leave PrP*^*Sc*^
*aggregates in ADAM10 cKO*.

We agree with this reviewer that the discrimination of infectious species, toxic entities and protease resistant PrP^Sc^ in the framework of prion diseases is an important issue with relevance to the whole field. Although we believe that our observation of an uncoupling of PrP^Sc^ and infectivity titers in forebrain samples at 95 dpi might contribute to this ongoing discussion, we again have to admit that our experimental design did not consider the requested experiment (see response to point 1 and 4; Figure 3). Nevertheless we did not find evidence for existence of atypical PrP fragments that could correspond to infectivity and explain this uncoupling observed in forebrain samples (as shown in new Figure 6—figure supplement 1 and mentioned in our response to Reviewer #2, point 7).

*6) Discussion: The authors state that PrP “is a receptor mediating neurotoxicity in common neurodegenerative proteinopathies such as Alzheimer’s disease”. This is a highly controversial contention propagated by Dr. Strittmatter, and there is more than half-dozen papers out there showing that this idea is, at best, limited to certain animal models. It is the authors' privilege to choose what they wish to believe, and I understand that the charitable interpretation of the Strittmatter paper might add to the perceived importance of the current study, but in fairness they should at least mention the evidence against PrP*^*C*^
*being the Aβ receptor.*

We agree with this reviewer that there is considerable controversy regarding the suggested role of PrP^C^ as a receptor for Aβ. This controversy becomes even stronger with regard to the downstream effects of this interaction and their contribution to synaptic impairment and neuronal loss in Alzheimer's disease (AD). For this reason we have now mentioned this scientific dispute and referenced three studies and a close-up article covering this controversy in the Introduction (as we could not find the statement the reviewer referred to in our Discussion):

“Despite some degree of controversy regarding this receptor function and its relevance in AD (9; 14; 22; 48), binding of pathogenic oligomers […]” (second paragraph).

In addition, we have toned down statements in the Abstract (“Moreover, it has been suggested as a receptor mediating neurotoxicity in common neurodegenerative proteinopathies such as Alzheimer’s disease”) and in the conclusion (“In view of a potential role of PrP^C^ in Alzheimer's disease…”; last paragraph of the Discussion section). Given that this issue is not of central relevance for our study, we decided to avoid a detailed discussion in the manuscript. However, to the best of our knowledge the majority of studies that have tackled this topic in the meantime (including a study from our own group investigating binding of Aβ to PrP^C^ in AD brains; [30]) rather support the role of PrP^C^ as a specific binding partner of Aβ.

*7) One problematic point is the following: It is stated that ADAM10 cKO mice show massive neurodegeneration in the forebrain, but not in the cerebellum and brainstem, while controls including* tga20 *show diffuse neurodegeneration. If this is really true, how can one explain completely identical clinical phenotypes?*

We are thankful for this question as it gives us the opportunity to clarify that we did not describe “completely identical phenotypes”. In fact, *tg*a*20* mice show a much more rapid progression of clinical signs (33). While most of those signs were indeed comparable between ADAM10 cKO, wild-type control and *tg*a*20* mice, we rarely observed kyphosis and ungroomed fur in the latter. In contrast to the other experimental groups, almost all *tg*a*20* mice developed a hind leg paresis. This might be related to the cerebellar pathology found in these mice (although this remains speculative). Nevertheless, we have added the data for *tg*a*20* mice in Table 1 and comparisons between genotypes and brain regions are summarized in Figure 3 (as requested by Reviewer #2).

*8) As for*
Figure 5*, the total amount of calpain is most certainly irrelevant to neurodegeneration, since calpains are mostly inactive in the absence of raised intracellular calcium. What matters is the amount of activated calpains. This can be assessed by investigating the amount of cleavage of a calpain substrate, such as spectrin (alpha-fodrin). The mere detection of calpains by Western blot is irrelevant. The whole argument about calpains needs to be revisited by activity measurements, or else it should be deleted in toto. This becomes painfully evident by the authors' finding that on the one hand there is prion toxicity (allegedly through calpain) with profound neurodegeneration in the forebrain of ADAM10 cKO, but on the other hand the significantly faster disease in* tga20 *mice goes along with significantly less calpain Also, a comparison of overall (not only forebrain) calpain activity levels to mock inoculated animals could help.*

As mentioned above, we have performed additional experiments and changed the manuscript text to address the points raised here.

Importantly we showed that increase of calpain expression in ADAM10 cKO (Figure 6) is specifically seen in prion disease since we could not detect differences in calpain level between non-infected ADAM10 cKO and controls. Thus, although calpain activity is of course the most relevant readout these specific differences deserve to be mentioned.

We used prion-infected mice in most experiments and their tissues have to be handled in a biosafety level 3* laboratory due to governmental regulation. Given this, sophisticated measurements to assess calpain activity (Ca^2+^ imaging or detection of fluorogenic substrates) were not possible. As mentioned above and described in Figure 3 all biochemical analyses had to be done on forebrain because the other reasons were used for morphological analysis.

Nevertheless, we followed the advice of this reviewer and detected spectrin. Additionally we decided to detect p35, another calpain substrate implicated in neurodegeneration (58). Full-length versions of both proteins appeared reduced in ADAM10 cKO. This decrease was not accompanied by increased levels of cleavage products, which may be due to technical reasons.

Thus, this point remains to be further investigated. We have included these data in the Results section (seventeenth paragraph), in the new Figure 6—figure supplement 2, and we have toned down our statements in the Abstract, Discussion and in the description of our model (Figure 9):

“This is specific for prion disease as no differences were detected in non-prion infected ADAM10 cKO mice (mean: 0.90 ±0.26 SEM; n=4) when compared to age-matched wild-type controls (set to 1 ±0.19; n=5; *p*=0.7) (Figure 6—figure supplement 2). In prion disease, we observed a reduction of the described calpain substrates spectrin (32) and neuron-specific activator p35 (58) suggesting that increased expression in ADAM10 cKO mice correlated with activity of calpain (Figure 6—figure supplement 2). However, it should be noted that we failed to detect differences in specific cleavage products of spectrin and p35.”

*9) When mentioning the possible inhibition of PrP*^*Sc*^
*formation by shedded PrP*^*C*^*, it would be desirable to prove this statement with PMCA + shed PrP*^*C*^.

We agree that a PMCA-based assessment of the role of shed PrP^C^ on PrP^Sc^ formation would be a desirable addition to the paper. However, obtaining quantities of physiologically shed PrP^C^ (and its purification) necessary for this method is not trivial. Usage of recombinant PrP has other disadvantages and does not fully recapitulate the in vivo situation (49). This reference has now been added in the Discussion part covering the inhibition of PrP^Sc^ formation by anchorless PrP versions (sixth paragraph). Thus, we have decided to address this in more detail in a separate study.

*10) In*
Figure 1*, POM2 is named as the applied antibody, while the Methods section describes POM1 for this purpose*.

This mismatch has been corrected. As now stated in the figure legend for Figure 1 and Methods section, POM2 was used for detection of shed PrP^C^ in this experiment.